# TEST TIME TRAINING FOR SUPERVISED CAUSAL LEARNING

## ABSTRACT

Supervised Causal Learning (SCL) has shown promise in causal discovery by framing it as a supervised learning problem. However, it suffers from significant out-of-distribution generalization challenges. We reveals three fundamental limitations of previous SCL practices: fragility to distribution shifts, failure in compositional generalization, and a significant performance gap between synthetic benchmarks and real-world data, collectively questioning its real-world applicability. To address this, we propose Test-Time Training for Supervised Causal Learning (TTT-SCL), a novel framework that dynamically generates training data explicitly aligned with any specific test instance. We find that the similarity between training and test data can be implicitly captured through distributional alignment, which we operationalize via a proposed Alignment of Distribution (AD) metric. To prevent degenerate solutions and enforce causal minimality, we incorporate sparsity constraints into the optimization. Building on this foundation, we introduce Test-time Aligned Causal Training with Informed Construction (TACTIC), the first instantiation of TTT-SCL, which jointly optimizes AD and sparsity via stochastic graph refinement to dynamically generate aligned training data. Experiments on synthetic benchmarks, pseudo-real and real-world dataset demonstrate that TACTIC significantly outperforms existing SCL and traditional causal discovery methods.

## 1 INTRODUCTION

Causal discovery aims to infer causal relationships from observational data (Pearl, 2009; Spirtes et al., 2000). While traditionally approached as an unsupervised problem (Fig. 1 (a)), Supervised Causal Learning (SCL) has recently emerged as a promising alternative (Dai et al., 2023; Lorch et al., 2022; Ke et al., 2022; Zhang et al., 2025). SCL treats causal discovery as a supervised learning task: a model is trained on a set of synthetic causal instances, each comprising a causal graph and its corresponding sampled dataset, and learns to map datasets to their underlying causal graphs (Fig. 1 (b)).

A pivotal factor for SCL's success is the design of its training set. What properties should these synthetic training set possess to ensure the model performs well on a real-world, unseen test instance? Two complementary principles guide this design: **diversity** and **concentration**. Diversity seeks broad coverage of possible causal models by varying key components such as graph structures, mechanisms, and noise characteristics, thereby encouraging generalization. Concentration, in contrast, aims to align the training set closely with the specific characteristics of the test domain of interest.

Current SCL methods heavily prioritize diversity, pre-training on large, static training sets generated from wide-ranging synthetic distributions (Lorch et al., 2022; Ke et al., 2022). However, we demonstrate that this paradigm suffers from critical limitations. Through systematic experiments, we find that these models are **fragile under distribution shifts**, showing significant performance degradation when the test instance differs from the training set in graph structure, causal mechanism, or noise distribution. More fundamentally, they **fail to generalize compositionally**; even when trained on all individual components, they cannot handle novel combinations of these components. Critically, this results in a pronounced generalization gap, where **strong performance**

**on synthetic benchmarks fails to translate to real-world data**, thereby questioning the practical utility of existing SCL approaches.

These findings motivate a paradigm shift from diversity to concentration. We argue that robust generalization requires moving beyond a single, fixed training set and instead dynamically adapting to each test instance. To this end, we introduce a Test-Time Training for Supervised Causal Learning (TTT-SCL) framework. The core idea is that for a given test dataset, we dynamically generate a new, customized training set that is explicitly aligned with its distribution, train a specialized SCL model, and utilize this model to infer the causal graph (Fig. 1 (c)).

The central challenge of TTT-SCL is ensuring this alignment. Our key insight is that the similarity between a candidate causal instance and the true, unknown instance can be captured implicitly through the similarity of their data distributions. We operationalize this idea via a proposed Alignment of Distribution (AD) metric. To prevent trivial solutions and enforce causal minimality, we incorporate a sparsity constraint. This combination allows us to efficiently search for high-quality, test-aligned causal instances for training.

Building on this, we propose Test-time Aligned Causal Training with Informed Construction (TAC-TIC), **the first concrete instantiation of the TTT-SCL framework**. TACTIC performs a stochastic search in the space of causal graphs, jointly optimizing for distributional alignment and sparsity to construct an effective training set for each test instance. Experiments on synthetic, pseudo-real, and real-world data show that TACTIC significantly outperforms existing SCL and traditional causal discovery methods.

Our main contributions are:

- We reveals three fundamental limitations of static SCL pre-training: fragility to distribution shifts, failure in compositional generalization, and a significant performance gap between synthetic benchmarks and real-world data, collectively questioning its real-world applicability.

- We introduce the TTT-SCL framework, enabling dynamic generation of aligned training data at test time. This includes the formulation of AD as a tractable metric for similarity via distributional alignment, and a sparsity constraint that ensures causal minimality and avoids degenerate graphs.

- We propose TACTIC, the first concrete method under TTT-SCL. TACTIC dynamically constructs effective training datasets tailored to each test instance, achieving excellent performance across both synthetic, pseudo-real and real-world datasets.

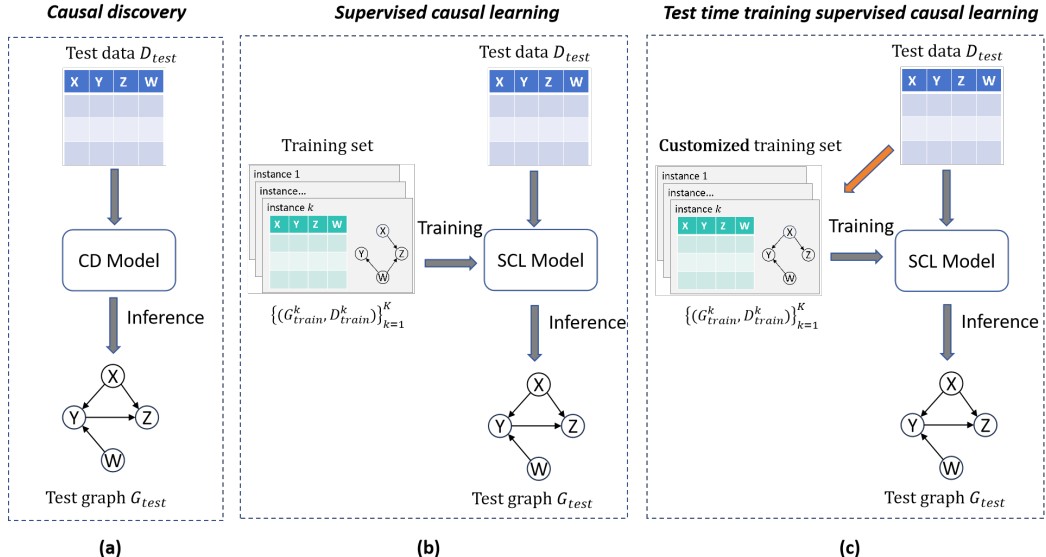

Figure 1: Test time training supervised causal learning compare with causal discovery (unsupervised causal learning) and supervised causal learning.

## 2 BACKGROUND

We begin by formalizing the core components of causal learning. A Structural Causal Model (SCM) consists of three key elements: causal graph, causal mechanisms, and noise distributions (Pearl, 2009; Peters et al., 2017). Specifically:

- Causal Graph: Let $G = (V, E)$ be a Directed Acyclic Graph (DAG) with vertex set $V = \{X_1, \ldots, X_d\}$ and edge set $E \subseteq V \times V$, where $d$ is the number of variables. The adjacency matrix $A \in \{0, 1\}^{d \times d}$ encodes edge relationships where $A_{ij} = 1$ iff $X_i \to X_j \in E$.

- **Causal mechanisms and noise:** Each variable $X_i$ is generated by a causal mechanism and exogenous noise, following the Structural Causal Model (SCM) framework (Pearl, 2009). The data-generating process is characterized by the structural equations:

$$X_i := f_i(\mathbf{Pa}_G(X_i), \varepsilon_i), \tag{1}$$

where $\mathbf{Pa}_G(X_i)$ denotes parents of $X_i$ in $G$, $f_i : \mathbb{R}^{|\mathbf{Pa}_G(X_i)|} \times \mathcal{E}_i \to \mathbb{R}$ is the causal mechanism, and $\varepsilon_i \sim P_{\varepsilon_i}$ is exogenous noise from distribution $P_{\varepsilon_i}$. The full SCM is thus characterized by the tuple $(G, \{f_i\}_{i=1}^d, \{\varepsilon_i\}_{i=1}^d)$, which comprehensively captures the causal structure, functional relationships, and exogenous noise.

In supervised causal learning, we work with **causal instances**. A causal instance is defined by a graph $G$ and a dataset $D$ containing $n$ observations $\{\mathbf{x}^{(1)}, \ldots, \mathbf{x}^{(n)}\} \in \mathbb{R}^{n \times d}$, generated from the SCM $(G, \{f_i\}_{i=1}^d, \{\varepsilon_i\}_{i=1}^d)$. The **training set** comprises $K$ such instances, denoted as $\{(D_{train}^k, G_{train}^k)\}_{k=1}^K$, where each $D_{train}^k$ is generated from its corresponding $G_{train}^k$. Similarly, at test time, we are given a single **test instance** $(D_{test}, G_{test})$, where $D_{test}$ is observed but $G_{test}$ is unknown. To avoid notation clutter, we adopt the following conventions: indices $i, j$ refer to variable/node indices within a graph, and subscripts "train" and "test" distinguish between training and test entities.

Causal discovery aims to estimate the causal graph $G_{test}$ from $D_{test}$ using a model or algorithm $M$. Supervised causal learning (SCL) frames this as a supervised learning problem, where a model (typically a neural network) is trained on synthetic causal instances to learn a mapping from observational data to graph structures. Formally, the SCL objective is to learn:

$$\mathcal{M} : \mathbb{R}^{n \times d} \to \{0, 1\}^{d \times d}, \tag{2}$$

which maps an input data matrix (e.g., $D_{test}$) to an output adjacency matrix (representing $G_{test}$). The model is trained on synthetic pairs $\{(D_{train}^k, G_{train}^k)\}_{k=1}^K$.

Previous SCL methods rely on training with **synthetic data**, where the generative distribution is explicitly controlled along three dimensions consistent with the SCM framework: graph structure, causal mechanisms, and noise distributions (Lorch et al., 2022; Ke et al., 2022; Froehlich & Koeppl, 2024). Typically, graphs are sampled from random graph models (e.g., Erdős–Rényi (Gilbert, 1959), Scale-Free (Barabási, 2009); mechanisms are chosen from a limited set of function classes (e.g., Linear, Random Fourier features (Rahimi & Recht, 2007)); and noise is drawn from parametric families (e.g., Gaussian, Uniform).

## 3 OUT-OF-DISTRIBUTION CHALLENGES FOR SCL

Out-of-distribution generalization has long been a challenge in machine learning, and we will show that it poses particularly severe implications for SCL. Unlike conventional ML domains where real-world training data is often available, SCL faces a fundamental constraint: causal graphs are rarely available for real-world datasets. This forces SCL methods to rely largely on synthetic training data, making the bridge between synthetic simulation and real-world application the primary bottleneck for SCL.

Current SCL models are typically evaluated under constrained synthetic shifts, for instance, training and testing on the same mechanism type with slightly different parameter ranges. While such evaluations demonstrate robustness to mild parametric variations, they represent a weak form of generalization that remains within synthetic data distributions. These approaches cover only narrow

mechanism families, while real-world causal relationships may involve complex, unmodeled functional forms. When test mechanisms fall outside the convex hull of training mechanisms, structural diversity alone cannot guarantee accurate estimation.

We point out three issues in previous SCL practices that collectively undermine their real-world applicability. First, these models are vulnerable to distribution shifts, exhibiting performance degradation when test distributions differ categorically from training in graph structure, mechanisms, or noise **(Issue 1)** . Second, they fail in compositional generalization, as models trained on diverse components cannot handle novel combinations of them, suggesting mere memorization of training configurations rather than learning modular causal representations **(Issue 2)**. Third, and most critically, they show divergent generalization patterns where strong performance on synthetic benchmarks fails to translate to real-world data, revealing a fundamental overfitting to the synthetic domain **(Issue 3)**. We use a series of experiments to illustrate these issues.

## 3.1 EXPERIMENT SETUP

**Datasets.** To comprehensively evaluate generalization, we use both synthetic benchmarks, pseudo-real and real-world dataset.

- **Synthetic data:** We generate test instances from a factorial combination of mechanism, graph and noise distrbution. We use three mechanism classes: Linear, Random Fourier Features (RFF) (Rahimi & Recht, 2007), and Chebyshev polynomials (Froehlich & Koeppl, 2024). We use two random graph models: Erdos-Renyi (ER) and Scale-Free (SF) (Gilbert, 1959; Barabási, 2009). Gaussian noise is used for RFF and Chebyshev mechanisms, while Uniform noise is used for Linear mechanisms to ensure identifiability. This yields six test settings: RFF_G_ER, RFF_G_SF, Linear_U_ER, Linear_U_SF, Chebyshev_G_ER, and Chebyshev_G_SF.

- **Real-world data:** We use the Sachs dataset (Sachs et al., 2005), a well-established benchmark in causal discovery. It contains 853 measurements of 11 proteins and a consensus causal graph derived from biological knowledge.

- **Pseudo-real data**: We also incorporate pseudo-real datasets generated by the SynTReN generator (Van den Bulcke et al., 2006). This generator is specifically designed to simulate synthetic transcriptional regulatory networks with biologically plausible structures and parameters, producing gene expression data that closely resembles experimental microarray data.

**Model & Training setting.**: We mainly use the AVICI as the model backbone (Lorch et al., 2022), a DNN-based architecture which is currently widely followed by the community and open-sourced. Results with other backbones are consistent and shown in Appendix C. To assess different generalization aspects, we compare several training settings, more detailed configurations can be found in the appendix B:

- **i.i.d**: The training data and test data are exactly the same distribution.

- **Graph/Noise/Mechanism shift**: The mechanism/graph/noise of the training data is different from that of the test data, but the other two distributions are the same.

- **Component-mixed**: This training setup contains all individual components (mechanisms, graph types, noise distributions) seen in isolation during training, but crucially excludes the specific combinations present in the test instances. This tests whether the model can perform *compositional generalization* by recombining learned components, rather than merely memorizing training configurations.

- **AVICI (scm-v0)**: This model was trained on SCM data simulated from a large variety of graph models with up to 100 nodes, both linear and nonlinear causal mechanisms, and homogeneous and heterogeneous additive noise from Gaussian, Laplace, and Cauchy distributions. It can be considered one of the strongest model of open source under the SCL paradigm. (https://github.com/larslorch/avici)

## 3.2 LIMITATIONS OF CURRENT SCL PARADIGMS

Our experimental results validate the three issues outlined above, collectively exposing the limitations of static pre-training in SCL.

**Issue 1**. The results in Fig 2 demonstrate that distribution shifts across all three dimensions (graph structure, causal mechanism, and noise distribution) significantly degrade SCL performance. Models struggle when the test-time graph structure ("Graph shift" compared to "i.i.d"), causal mechanism ("Mechanism shift" compared to "i.i.d"), or noise distribution ("Noise shift" compared to "i.i.d") differs categorically from those seen during training. While performance drops are observed in all cases, "Mechanism shifts" emerge as particularly damaging, underscoring the profound impact of the underlying mechanism functional form on model generalization.

**Issue 2**. Even when trained on data containing all individual components, the model still exhibits performance drop on unseen combinations of these components, as seen when comparing "Component-mixed" to "i.i.d" in Fig 2. This compositional failure indicates that SCL models memorize specific $(G, f, \varepsilon)$ configurations rather than learning a modular understanding of causal factors.

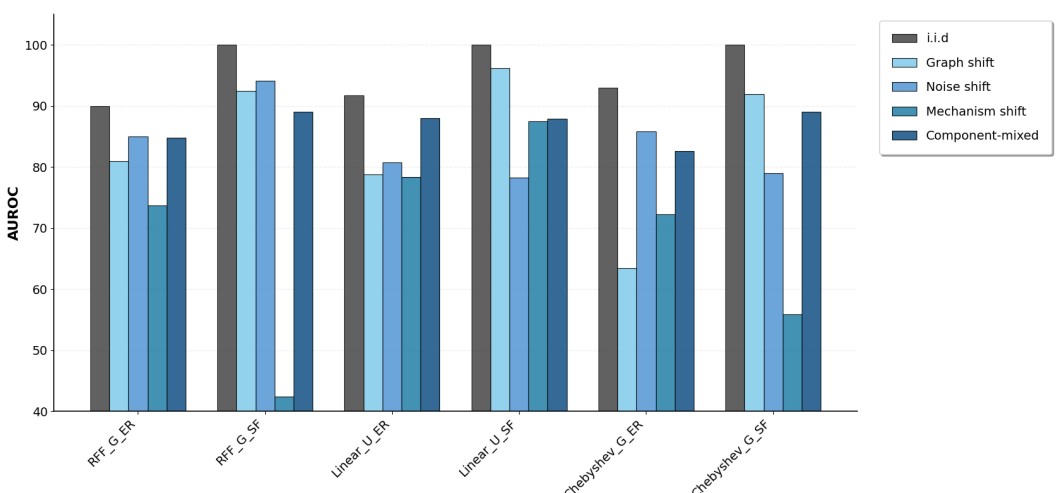

Figure 2: Two fundamental limitations of static SCL: fragility to distribution shifts and failure in compositional generalization.

**Issue 3**. The results in Table 1 question the value of synthetic benchmarks by demonstrating that strong synthetic performance fails to guarantee effectiveness on real-world data. Here, we merge the dimensions of the graph and analyze more from the perspective of the mechanism. While AVICI (scm-v0) excels on synthetic data similar to its training distribution (e.g., RFF_G, 97.8), its performance collapses on the real-world Sachs dataset (62.3). In contrast, traditional methods like PC maintain consistent, albeit lower, performance across domains. This divergence reveals that SCL models overfit to the artifacts of their synthetic training set, lacking the cross-domain consistency required for real-world applicability.

Table 1: Divergent generalization patterns. Strong synthetic performance does not guarantee effectiveness on real-world data. Results are presented as AUROC (standard deviation).

|  | RFF_G | Linear_U | Chebyshev_G | Sachs | Syntren |
|---|---|---|---|---|---|
| PC | 61.1 (4.9) | 60.9 (4.7) | 59.8 (6.6) | 67.1 | 58.1 |
| AVICI (scm-v0) | 97.8 (1.3) | 75.6 (13.8) | 81.7 (10.5) | 62.3 | 65.4 |

In summary, the dual failure of fragility under distribution shifts and inconsistency across domains fundamentally undermines the static pre-training paradigm. These limitations are not artifacts of a specific architecture, as validated by consistent failure patterns using the SiCL backbone (Appendix C). The results compellingly argue that robust causal discovery requires a shift from static, diversity-seeking pre-training to dynamic, test-time adaptation.

## 4 TEST-TIME TRAINING FOR SUPERVISED CAUSAL LEARNING

From the perspective of concentration, there remains an opportunity for SCL to overcome the limitations of static pretraining. We introduce the Test-Time Training for Supervised Causal Learning (TTT-SCL) framework, representing a paradigm shift from seeking universal diversity to generating targeted concentration, as shown in Fig 1.

Essentially, our TTT-SCL framework is general and learning target-independent. However, for intuitiveness and applicability, we set the learning target to causal graphs (DAG) in this paper. Therefore, our method is applicable to any assumption that guarantees the identification of the underlying causal graphs, such as LiNGAM (Shimizu et al., 2006), the nonlinear Additive Noise Model (Hoyer et al., 2008), or the Post-NonLinear model (Zhang & Hyvärinen, 2009). Since we aims to learn a mapping from the test dataset $D_{test}$ to its corresponding causal graph $G_{test}$, it is essential for the dataset $D_{train}^k$ in the training set to be as similar as possible to the test dataset $D_{test}$. Given that the distribution of a dataset depends on the underlying causal graphs and their parameterization, the problem can be transformed into finding, among the candidate graphs $G_{train}^k$ those that yield data distributions similar to the test dataset $D_{test}$. This objective can be further divided into the following two sub-problems:

- **Quantifying similarity.** What metric can we use to quantify "similarity" between a candidate graph $G_{train}^k$ and the test dataset $D_{test}$?
- **Searching effectively.** Given the intractability of brute-force search over the DAG space, how can we design a practical search procedure to identify promising candidates?

### 4.1 QUANTIFYING SIMILARITY: THE ALIGNMENT OF DISTRIBUTION

A natural way to connect candidate graphs $G_{train}^k$ with the test dataset $D_{test}$ is through Structure-Induced Mechanism (SIM). SIM directly operationalizes how a graph explains data: given a candidate graph $G_{train}^k$, we regress the corresponding mechanisms from the observed $D_{test}$, and then forward-sample synthetic data $D_{train}^k$. If the generated distribution is close to $D_{test}$, this indicates that $G_{train}^k$ is a good approximation of the true graph $G_{test}$. In this sense, SIM provides a practical bridge from structural hypotheses to observable distributional alignment, making it possible to evaluate candidate graphs by how well they reproduce the test distribution.

This motivates the need for a metric of alignment between a candidate training graph and the test data. Such a metric, which we denote as Alignment of Distribution (AD), should satisfy structure and mechanism similarity. While there are many ways to implement AD as discussed in Appendix A, in the main text we use the implementation based on likelihood:

$$AD(G_{train}^k, D_{test}) = \frac{1}{d} \sum_{i=1}^{d} \left[ \log p\left(X_i \mid f_i^k\right) \right], \tag{3}$$

where $f_i^k$ is the fitting function of $X_i$ according $\mathbf{Pa}_{train}^k(X_i)$ based on $G_{train}^k$ and $D_{test}$ by SIM.

This formulation is attractive because likelihood inherently combines both structure and mechanism aspects. Changing the graph structure alters the conditioning set $\mathbf{Pa}_{train}^k(X_i)$, directly modifying the conditional distributions being estimated. Changing the mechanisms alters the functional mapping $f_i^k$, thereby changing the probability assigned to the observed data. As a result, the likelihood score simultaneously reflects structural correctness and mechanistic fidelity, and thus serves as a principled measure of distributional alignment between candidate training graphs and the test data.

**Enforcing Causal Minimality with Sparsity Constraints.** However, optimizing AD alone can lead to degenerate dense solutions that fit distributions without respecting causal minimality. To counteract this, we incorporate the principle of causal minimality by adding a sparsity penalty term based on the $L_0$ norm of the adjacency matrix $A_G$:

$$\text{Sparsity}(G) = ||A_G||_0. \tag{4}$$

**The Joint Optimization Score.** By combining these two components, we form a unified score function to evaluate any candidate training graph:

$$score(G) = \text{AD}(G, D_{\text{test}}) - \lambda \cdot \text{Sparsity}(G). \tag{5}$$

where $\lambda$ is a hyperparameter balancing the trade-off. This score serves as the central optimization target for generating high-quality training data within the TTT-SCL framework.

## 4.2 TACTIC: EFFICIENT SEARCH IN THE GRAPH SPACE

Exhaustively searching the entire DAG space is intractable, and theoretical results confirm that finding the exact $G_{test}$ is essentially impossible. Nevertheless, this does not imply that the problem is hopeless. In practice, good initializations combined with guided refinement can yield graphs that are close enough to $G_{test}$ to support effective training. We instantiate this idea with TACTIC (Test-time Aligned Causal Training with Informed Construction), a concrete implementation of our TTT-SCL framework. TACTIC proceeds in three stages:

1. **Seed Initialization.** We start from an initial graph $G_{seed}$, obtained either by (i) applying a traditional causal discovery method (e.g., PC, NOTEARS) on $D_{test}$, or (ii) sampling a random DAG. This provides a useful prior rather than searching from scratch.

2. **Stochastic Graph Refinement.** From the seed, we iteratively propose local modifications to the graph (edge additions, deletions, or reversals) while maintaining the DAG constraint. Each candidate $G_{k+1}$ is evaluated using the joint score function $score(G)$ as Formula (5) and accepted with probability proportional to its score. This stochastic refinement process ensures that search is efficient and directed, guided by AD and sparsity rather than random exploration.

3. **Training Data Generation.** For the final refined graph set $\{G_{train}^k\}_{k=1}^K$, we regress mechanisms via SIM, forward-sample synthetic datasets $\{D_{train}^k\}_{k=1}^K$, and assemble them into a customized training set. We set the noise distribution to a standard Gaussian distribution $\mathcal{N}(0,1)$ by default. An SCL model is then trained on this set and applied to infer $G_{test}$.

By combining AD, sparsity, and practical heuristics (initialization + stochastic refinement), TACTIC realizes an efficient and directed approach to searching the graph space at test time, as shown in Fig 3. Complexity analysis and runtime variation with the number of nodes are detailed in Appendix F.

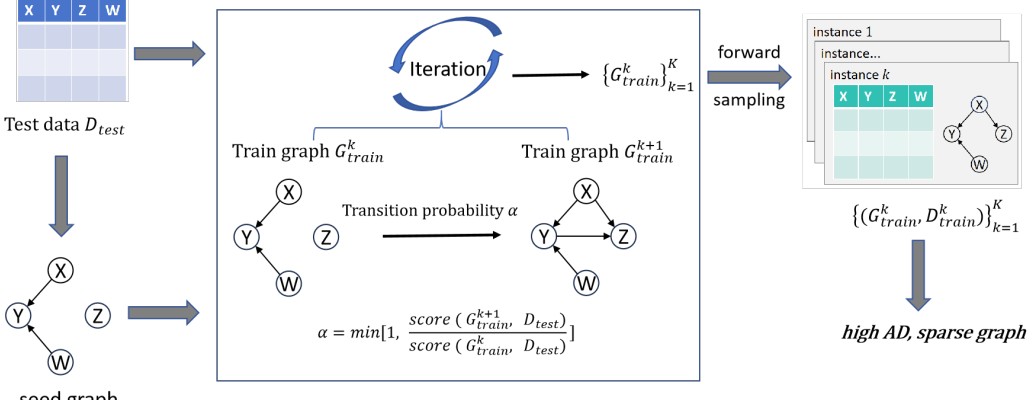

Figure 3: Workflow of TACTIC

## 4.3 THE PERFORMANCE OF TACTIC

In this subsection, we compare the performance of TACTIC with multiple baseline methods on various synthetic data, pseudo-real data and real data. These datasets are consistent with the content of Section 3.1.

**Baselines:** We compare against traditional causal discovery methods PC (Spirtes et al., 2000), GES (Chickering, 2002), NOTEARS (Zheng et al., 2018), RESIT (Peters et al., 2014), SCORE (Rolland et al., 2022), NoGAM (Montagna et al., 2023) and AVICI (Lorch et al., 2022), a DNN-based SCL method which is currently widely followed by the community and open source. We use the open-source pre-trained AVICI (scm-v0) model, which is trained on a vast mixture of synthetic data and represents the strongest publicly available SCL baseline.

**Our Method (TACTIC):** For our TTT-SCL approach, we set the number of dynamically generated training graphs to $K = 200$. The number of variables $d$ is 10 for synthetic data, 11 for Sachs and 20 for Syntren. The observation $n$ for each generated dataset matches that of the test data. We evaluate two variants of our method: TACTIC (random) which initializes the seed graph with a random DAG, and TACTIC (Notears) which uses a graph estimated from $D_{test}$ by the NOTEARS algorithm as a smarter starting point.

**Evaluation metrics:** We use multiple metrics to evaluate the predicted graphs, including Area Under the Receiver Operating Curve (AUROC), Area Under the Precision-Recall Curve (AUPRC), F1 score and Accuracy (ACC). In the main text, we primarily report **AUROC** for edge prediction to succinctly explore the impact of training data quality on model performance. Results based on other metrics are provided in Appendix D.

Table 2: TACTIC performance on synthetic, real and pseudo-real datasets. Results are presented as AUROC (standard deviation).

|                    | RFF_G        | Linear_U      | Chebyshev_G   | Sachs | Syntren |
|--------------------|--------------|---------------|---------------|-------|---------|
| PC                 | 61.1 (4.9)   | 60.9 (4.7)    | 59.8 (6.6)    | 67.1  | 58.1    |
| GES                | 66.0 (10.6)  | 69.0 (10.8)   | 59.6 (5.9)    | 61.8  | 36.8    |
| Notears            | 80.5 (4.0)   | 82.0 (4.6)    | 52.2 (3.5)    | 61.8  | 49.8    |
| RESIT              | 54.3 (5.4)   | 54.1 (5.2)    | 49.8 (4.7)    | 62.3  | 64.6    |
| SCORE              | 86.9 (3.2)   | 82.2 (18.7)   | 69.2 (7.6)    | 64.9  | 41.0    |
| NoGAM              | 87.6 (2.9)   | 79.2 (18.6)   | 72.3 (6.4)    | 64.9  | 41.0    |
| AVICI (scm-v0)     | **97.8** (1.3) | 75.6 (13.8) | 81.7 (10.5)   | 62.3  | 65.4    |
| TACTIC (random)    | 88.4 (7.0)   | 82.3 (7.0)    | 79.6 (6.7)    | 58.6  | 72.0    |
| TACTIC (Notears)   | 91.8 (3.1)   | **86.3** (4.4) | **83.0** (8.7) | **78.9** | **80.1** |

The results are summarized in Table 2. Overall, TACTIC demonstrates robust and highly competitive performance. The pre-trained AVICI (scm-v0) model achieves optimal performance on the RFF_G datasets, as it was explicitly trained on this distribution. TACTIC's performance on RFF_G is slightly lower but remains strong, indicating its ability to approximate even in-distribution performance without prior exposure. Crucially, TACTIC achieves state-of-the-art performance on all other datasets, including Linear_U, Chebyshev_G, real-world Sachs, and pseudo-real Syntren dataset. This confirms that TACTIC excels in the most challenging and realistic scenarios involving distribution shifts, where static pre-training fails. Furthermore, the TACTIC (Notears) variant consistently outperforms TACTIC (random), demonstrating that a reasonable initial graph from a traditional method provides a valuable prior for the optimization. The strong performance of both variants confirms the robustness of our core approach. These conclusions hold consistently across multiple evaluation metrics, as demonstrated in Appendix D, where TACTIC maintains superior performance in ACC, F1-score, and AUPRC under various distribution shifts. Appendix G presents results on four additional established benchmark causal graphs from the bnlearn repository (Asia, Cancer, Earthquake, and Survey), demonstrating TACTIC's robust performance across diverse real-world causal structures.

## 4.4 ABLATION STUDY

We design experiments to empirically validate how these two components contribute to the quality of the generated training data. We first ablate the sparsity term in the optimization objective to isolate its effect. We compare the full **TACTIC (Notears)** method against a variant, **TACTIC (Notears-s)**, where the sparsity penalty is removed ($\lambda = 0$), thus optimizing for AD alone. Results

in Table 3 show that removing the sparsity term leads to a consistent and significant performance drop across all test settings. These dense graphs achieve high AD by introducing spurious edges with negligible mechanisms, but they violate the causal minimality principle and thus constitute poor-quality training data for teaching the SCL model the correct causal structure.

Table 3: Ablation experiment of sparsity. Results are presented as AUROC (standard deviation).

|  | RFF_G | Linear_U | Chebyshev_G | Sachs | Syntren |
|---|---|---|---|---|---|
| TACTIC (Notears) | 91.8 (3.1) | 86.3 (4.4) | 83.0 (8.7) | 78.9 | 80.1 |
| TACTIC (Notears-s) | 86.8 (2.9) | 84.3 (7.9) | 69.7 (12.4) | 63.5 | 76.1 |

To further demonstrate the effectiveness of AD and the necessity of sparsity, the AD, sparsity, score of the training data obtained by different methods under different test data, as well as the AUROC on the test data were recorded in Appendix E. The results show that both AD and sparsity are indispensable and important elements, and they have certain indicative significance for performance.

To clearly distinguish our approach from classical score-based methods, we provide a detailed stage-wise analysis comparing three key outputs: the seed graph, the highest-scoring graph found during TACTIC's search, and the final SCL prediction. The consistent performance improvement across these stages demonstrates the added value of the supervised learning phase. Specifically, we conducted experiments comparing three different outputs across four test domains (RFF, Linear, Chebyshev, and Sachs) for detailed analysis:

1. **Seed graph**: Initial graph from proxy methods

2. **Highest-score graph**: Highest-score graph found during TACTIC's stochastic refinement

3. **Final output**: Graph predicted by the SCL model trained on TACTIC-generated data

Table 4: Performance comparison (AUROC) across different stages of TACTIC.

|  | RFF_G | Linear_U | Chebyshev_G | Sachs |
|---|---|---|---|---|
| Seed graph | 80.5 | 82.0 | 52.2 | 61.8 |
| Highest-score graph in TACTIC search | 88.9 | 80.1 | 75.8 | 66.6 |
| Final graph from trained SCL model | **91.8** | **86.3** | **83.0** | **78.9** |

The results in Table 4 clearly demonstrate the two-stage improvement of our approach:

- **1→2 (Search Improvement)**: The higher AUROC of the highest-score graph compared to the seed graph shows that TACTIC's stochastic refinement effectively improves graph quality through distributional alignment.

- **2→3 (Learning Improvement)**: The consistent and substantial performance gain of the final SCL output over the highest-score graph demonstrates the crucial advantage of our approach. While score-based methods would stop at the highest-scoring graph, TACTIC uses this graph to generate training data that enables an SCL model to learn more accurate causal relationships.

This two-stage process, where we optimize for training data quality rather than directly for the final graph, constitutes the fundamental distinction between TACTIC and classical score-based causal discovery.

## 5 RELATED WORKS

Causal discovery has a long history rooted in constraint-based methods (e.g., PC, FCI (Spirtes et al., 2000)), function-based methods (e.g., LiNGAM (Shimizu et al., 2006), ANM (Hoyer et al., 2008)) and score-based methods (e.g., GES (Chickering, 2002), NOTEARS (Zheng et al., 2018), DAG-GNN (Yu et al., 2019), GraN-DAG (Lachapelle et al., 2020)). These approaches operate unsupervised and infer causal graphs directly from observational data using statistical independencies, asymmetry assumptions or various scores. While principled, they often suffer from high sample

complexity, sensitivity to faithfulness violations, and limited scalability to high-dimensional settings.

Supervised Causal Learning (SCL) has recently emerged as a promising paradigm that approaches causal discovery as a supervised learning problem (Dai et al., 2023; Lorch et al., 2022; Ke et al., 2022). It trains a machine learning model to take observational data as input and output the causal graph or relations and leverage powerful models to learn mappings from data patterns to causal structures, instead of hand-crafted heuristics. The analysis of SCL can be conducted from the following three aspects:

**Model architecture.** Prior SCL methods employ diverse architectures to map datasets to graphs. For example, Ma et al. (2022) propose cascade classifiers that sequentially test conditional independencies by increasing the conditioning order. Dai et al. (2023) design architecture featurizes variable neighborhoods and classifies unshielded triples. Lorch et al. (2022), Ke et al. (2022), and Froehlich & Koeppl (2024) use the attention-based transformer that treats the data as a 3D tensor (observations × variables × features) and alternates self-attention over samples and variables. In addition, Zhang et al. (2025) propose pairwise attention to capture the node features and node-pair features.

**Target output representation.** SCL methods target different representations of causal relationships. Some methods learn only the undirected skeleton of the graph, e.g. Ma et al. (2022) aims to recover the full skeleton. Others focus on orienting local structures: for instance, Dai et al. (2023) takes as input the graph skeleton and classifies each unshielded triple as a v-structure or not, then orients edges accordingly. Ke et al. (2022)'s transformer outputs a full directed adjacency matrix via an autoregressive decoder over all node pairs, and Lorch et al. (2022)'s network similarly predicts edge probabilities between every ordered pair. Many methods only guarantee recovery up to Markov equivalence: for example, Zhang et al. (2025) train a model to output the skeleton and v-structure and Froehlich & Koeppl (2024) learns the moralized graphs.

**Training data strategy and test time training.** While recent work by Montagna et al. (2024) has also investigated SCL generalization challenges, they primarily attribute performance drops to unseen individual components (e.g., mechanisms) and suggest increased diversity in pre-training as the solution. In contrast, we identify a different and more fundamental limitation: *compositional generalization* failure. SCL models fail on novel combinations of seen components, revealing the intractability of exhaustive static pre-training and motivating our TTT-SCL paradigm.

Rather than simply scaling up pre-training data, TTT-SCL represents the first framework to introduce test-time training to supervised causal learning. While test-time adaptation has shown promise in general machine learning domains (Liang et al., 2025; Sun et al., 2020; Wang et al., 2020; Liu et al., 2021; Sinha et al., 2023), our work pioneers its application to causal discovery by generating targeted training data that is causally aligned with each test instance.

## 6 CONCLUSION

In this work, we identified fundamental limitations of static SCL paradigms, demonstrating their fragility under distribution shifts, failure in compositional generalization, and poor transfer from synthetic benchmarks to real-world data. To address these out-of-distribution generalization challenges, we introduced TTT-SCL, a paradigm-shifting framework that addresses the out-of-distribution generalization problem in supervised causal learning through test-time training of causally-aligned data. Our proposed AD metric, combined with sparsity constraints, provides a tractable and effective way to ensure causal similarity between training and test data. The TACTIC method, as an instantiation of TTT-SCL, dynamically generates high-quality training data tailored to each test instance, achieving good performance on both synthetic, pseudo-real and real-world datasets. Our theoretical and empirical results underscore the effectiveness of AD and necessity of sparsity. This work not only advances the field of supervised causal learning but also opens new avenues for robust and adaptive causal discovery in real-world settings.

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

# A  IMPLEMENTATION OF AD

In the main text, we propose the Alignment of Distribution (AD) metric as a core measure of causal similarity between the generated training data $D_{train}$ and the test instance $D_{test}$. While the likelihood-based implementation was used in our primary experiments, we provide alternative formulations here to accommodate different data distributions and modeling assumptions.

## A.1  $R^2$-BASED AD

For continuous variables under additive noise models, the coefficient of determination ($R^2$) provides an intuitive measure of goodness-of-fit for each causal mechanism:

$$AD_{R^2}(G_{train}, D_{test}) = \frac{1}{d} \sum_{i=1}^{d} \left[ \frac{1}{K} \sum_{k=1}^{K} R^2 \left( f_i^k(\mathbf{Pa}^k(X_i)), X_i \right) \right]$$

This value approaches 1 when the fitted mechanisms explain the variance in $D_{test}$ well, indicating strong alignment.

## A.2  NORMALIZED WASSERSTEIN DISTANCE-BASED AD

For multi-modal or heavy-tailed distributions, the Wasserstein distance offers a robust metric for comparing empirical distributions. We define a *Normalized Wasserstein Distance (NWD)* based AD metric as follows:

For a given variable $X_i$ and a candidate graph $G^k$ with its fitted mechanism $f_i^k$, we compute:

$$\text{NWD}(f_i^k, G^k, D^{test}) := 1 - \frac{W_1 \left( \{x_i\}, \{f_i^k(\mathbf{Pa}^k(X_i))\} \right)}{\max(\mathcal{U}) - \min(\mathcal{U})}$$

where:

- $\{x_i\}$ are the observed values of $X_i$ in $D_{test}$.
- $\{f_i^k(\mathbf{Pa}^k(X_i))\}$ are the values generated by applying the fitted mechanism $f_i^k$ to the parent values in $D_{test}$.
- $W_1$ is the 1-Wasserstein distance (Earth Mover's Distance). For two equally sized, sorted collections of values $\{a^{(j)}\}$ and $\{b^{(j)}\}$, it is defined as:

$$W_1(\{a\}, \{b\}) = \frac{1}{n} \sum_{j=1}^{n} |a^{(j)} - b^{(j)}|$$

- $\mathcal{U} = \{x_i\} \cup \{f_i^k(\mathbf{Pa}^k(X_i))\}$ is the union of the observed and generated values for $X_i$.
- The denominator, $\max(\mathcal{U}) - \min(\mathcal{U})$, is the range of the combined set, used for normalization.

The resulting NWD value lies between 0 and 1, where 1 indicates a perfect match between the generated and observed distributions for that variable. The overall AD metric is then the average NWD across all variables and generated graphs:

$$AD_{\text{NWD}}(G_{train}, D_{test}) = \frac{1}{K} \sum_{k=1}^{K} \left[ \frac{1}{d} \sum_{i=1}^{d} \text{NWD}(f_i^k, G^k, D^{test}) \right]$$

## A.3  SELECTION GUIDANCE

The **likelihood-based** AD is most natural for probabilistic models and was used in our main experiments. The $R^2$-**based** AD is suitable for continuous variables under additive noise assumptions, often leading to computationally efficient and intuitive scores. The **NWD-based** AD is recommended for complex, non-Gaussian, or heavy-tailed distributions where likelihood or $R^2$ might be less informative or robust. The TTT-SCL framework is agnostic to the specific choice of AD metric, allowing users to select the most appropriate one for their domain.

## B   DETAILED CONFIGURATION OF TRAINING DATA

For all static SCL training setups evaluated(including i.i.d., Graph/Noise/Mechanism shift), we use a total of $K = 2,000$ synthetic training instances. Each instance contains $n = 200$ i.i.d. observations. The specific training settings for different test instances are as follows:

Table 5: Graph/Noise/Mechanism shift training data setting

|  | RFF_G_ER | RFF_G_SF | Linear_U_ER | Linear_U_SF | Chebyshev_G_ER | Chebyshev_G_SF |
|---|---|---|---|---|---|---|
| Graph shift | RFF_G_SF | RFF_G_ER | Linear_U_SF | Linear_U_ER | Chebysev_G_SF | Chebysev_G_ER |
| Noise shift | RFF_U_ER | RFF_U_SF | Linear_L_ER | Linear_L_SF | Chebysev_U_ER | Chebysev_U_SF |
| Mechanism shift | Chebysev_G_ER | Chebysev_G_SF | RFF_U_ER | RFF_U_SF | RFF_G_ER | RFF_G_SF |

In the Component-mixed setting, these 2,000 instances are uniformly distributed across the 6 mechanism_noise_graph combinations, resulting in approximately 330 instances per specific combination. The training data is a mixture of RFF_U_ER, RFF_U_SF, Linear_G_ER, Linear_G_SF, Chebysev_U_ER, and Chebysev_U_SF. This makes the model see all components, mechanism (RFF, Linear, Chebyshev), graph (ER, SF), noise (G, U), but not see the specific combination in the test instance, such as RFF_G_ER.

## C   CONSISTENCY ON OTHER MODEL BACKBONES

To further validate the generality of the TTT-SCL framework and the observed o.o.d generalization challenges across different model architectures, we conduct experiments using the Pairwise Attention from Zhang et al. (2025) (**SiCL**) as an alternative model backbone. Unlike the AVICI transformer used in the main experiments, which predicts a full directed adjacency matrix (DAG), SiCL incorporates pairwise attention mechanisms and is trained to predict the undirected skeleton and v-structures of the causal graph. This setup allows us to investigate whether the identified o.o.d failure patterns persist when using a fundamentally different architecture (with pairwise attention) and a different learning target (skeleton and v-structures instead of a full DAG), thereby testing the robustness of our conclusions.

### C.1   EXPERIMENTAL SETUP

Backbone Model is SiCL (Pairwise Attention Network) Zhang et al. (2025). Learning Target is Undirected graph skeleton. The training strategy for the static baseline models (i.i.d. and SiCL(mixed)) follows the same data generation procedures described in Section 5.1.1 of the main text, but the ground-truth labels are converted to the appropriate representation for SiCL (skeleton labels). Evaluation Metric is AUROC for edge presence in the predicted skeleton. OOD Settings is identical to those defined for Table 1 in the main text: *i.i.d.*, *Graph shift*, *Noise shift*, *Mechanism shift*. The *AVICI(mixed)* is replaced with *SiCL(mixed)*, respectively.

### C.2   RESULTS AND ANALYSIS

Table 6 presents the AUROC for skeleton discovery under different distribution shifts.Consistent with the findings in Fig 2 using the AVICI backbone, the SiCL backbone—which employs a fundamentally different pairwise attention architecture and learns undirected skeletons rather than full DAGs—exhibits the same pattern of out-of-distribution generalization failure. Under i.i.d. conditions, SiCL achieves perfect or near-perfect performance. However, significant performance degradation occurs across all types of distribution shifts, with mechanism shifts proving particularly damaging (e.g., dropping to 66.5 on RFF_G_SF and 58.4 on Chebyshev_G_SF). Critically, the SiCL(mixed) variant, while trained on data containing all individual distributional components (graph types, mechanisms, and noise distributions), still fails to generalize to novel combinations of these factors. This demonstrates that SCL models struggle with compositional generalization—they memorize specific configuration patterns rather than learning modular causal representations. These results demonstrate that the OOD generalization challenge is not specific to a particular model architecture or output representation, but represents a fundamental limitation of the static pre-training

paradigm in supervised causal learning. The consistent failure patterns across both transformer-based (AVICI) and pairwise-attention-based (SiCL) models strongly validate the need for test-time adaptation frameworks like TTT-SCL.

Table 6: OOD generalization performance for skeleton using the SiCL (Pairwise Attention) backbone.

|  | RFF_G_ER | RFF_G_SF | Linear_U_ER | Linear_U_SF | Chebysev_G_ER | Chebysev_G_SF |
|---|---|---|---|---|---|---|
| iid | 82.1(6.7) | 100.0(0.0) | 81.4(6.9) | 100.0(0.0) | 94.3(2.8) | 100.0(0.0) |
| Graph shift | 66.4(9.0) | 85.4(4.1) | 65.8(6.9) | 94.0(2.5) | 73.0(5.7) | 92.9(4.3) |
| Noise shift | 60.0(8.9) | 91.7(3.8) | 65.3(7.4) | 84.0(7.7) | 88.6(5.3) | 89.3(5.4) |
| Mechanism shift | 62.1(7.4) | 66.5(6.3) | 59.4(4.7) | 83.8(4.8) | 76.1(8.9) | 58.4(9.7) |
| SiCL(mixed) | 64.4(8.0) | 74.4(10.7) | 66.7(7.3) | 82.7(8.2) | 85.6(3.7) | 91.2(4.1) |

## D PERFORMANCE IN OTHER METRICS

In the main text, we primarily reported the AUROC for edge prediction to succinctly demonstrate the impact of training data quality on model performance. For a more comprehensive evaluation, we provide results on additional standard causal discovery metrics in this appendix:

- **Accuracy (ACC)**: The proportion of correctly predicted edge presence/absence across all possible edges. Higher is better. This metric can be viewed as a normalized version of the Structural Hamming Distance (SHD), where instead of counting the number of incorrect edges, it measures the proportion of correct edge predictions relative to the total possible edges.

- **F1-Score**: The harmonic mean of precision and recall for edge prediction. Higher is better.

- **Area Under the Precision-Recall Curve (AUPRC)**: Particularly informative under class imbalance (sparse graphs). Higher is better.

Table 7: Comprehensive evaluation across multiple datasets and metrics. Mean (standard deviation) over multiple runs are reported for synthetic data. Best results are in **bold**.

| Method | RFF_G | | | Linear_U | | | Chebyshev_G | | | Sachs | | | Syntren | | |
|---|---|---|---|---|---|---|---|---|---|---|---|---|---|---|---|
|  | ACC↑ | F1↑ | AUPRC↑ | ACC↑ | F1↑ | AUPRC↑ | ACC↑ | F1↑ | AUPRC↑ | ACC↑ | F1↑ | AUPRC↑ | ACC↑ | F1↑ | AUPRC↑ |
| PC | 75.6(4.2) | 39.5(9.7) | 37.5(6.3) | 74.0(4.4) | 40.4(8.4) | 37.2(6.2) | 73.7(5.1) | 37.4(12.6) | 36.8(7.8) | 84.2 | 45.7 | 30.1 | 84.75 | 16.43 | 6.89 |
| GES | 76.7(7.7) | 49.8(16.3) | 43.5(12.3) | 71.9(9.8) | 55.4(12.9) | 45.3(9.3) | 72.1(5.1) | 38.5(10.5) | 35.7(6.6) | 82.6 | 36.3 | 24.2 | 65.50 | 1.42 | 4.79 |
| NOTEARS | 86.6(3.3) | 73.2(6.2) | 64.1(7.4) | 89.1(2.9) | 76.6(7.3) | 69.7(8.6) | 72.3(2.7) | 14.5(8.7) | 29.8(3.3) | 82.6 | 36.3 | 24.2 | **94.75** | 0.00 | 5.00 |
| AVICI(scm-v0) | **93.1(1.6)** | **87.3(3.4)** | **94.9(3.1)** | 73.9(7.1) | 41.8(18.4) | 52.8(17.0) | 80.6(5.4) | 58.4(14.5) | 69.3(14.2) | 83.4 | 23.0 | 31.6 | 93.00 | 22.22 | 25.53 |
| TACTIC (random) | 83.2(5.3) | 72.8(9.4) | 68.8(10.8) | 75.9(6.1) | 59.8(11.8) | 56.2(11.5) | 75.5(7.0) | 56.6(10.8) | 60.0(10.0) | 68.5 | 24.0 | 24.5 | 72.50 | 16.66 | **53.91** |
| TACTIC (Notears) | 86.8(3.5) | 78.4(6.1) | 76.0(8.6) | **78.7(3.9)** | **65.4(8.0)** | **65.0(9.9)** | **77.1(6.7)** | **61.9(10.2)** | **66.0(16.3)** | **85.9** | **56.4** | **53.6** | 90.50 | **32.14** | 51.85 |

Table 7 presents the performance of all compared methods across three distinct synthetic data settings (RFF_G, Linear_U, and Chebyshev_G) and the real-world Sachs dataset. TACTIC (Notears) achieves highly competitive performance across all datasets and evaluation metrics (ACC, F1, AUPRC), demonstrating its robustness to distribution shifts. It consistently outperforms traditional methods (PC, GES, NOTEARS) and the strong pre-trained SCL baseline AVICI(scm-v0) on most settings, particularly on the challenging Chebyshev_G and real-world Sachs dataset. While AVICI(scm-v0) excels in the RFF_G setting it was trained on, its performance degrades significantly under mechanism shifts (Linear_U) and on real data, highlighting the limitation of static pre-training. The superior performance of TACTIC across multiple metrics confirms that its test-time training strategy generates high-quality, causally-aligned training data, leading to more accurate and reliable causal discovery.

## E MORE EXPERIMENTS ABOUT AD AND SPARSITY

The main text established the necessity of the sparsity constraint in the TACTIC optimization objective to prevent degenerate, overly dense solutions. This appendix provides further empirical evidence to dissect the roles of the AD metric and the sparsity constraint.

## E.1 THE ROLE OF AD AND SPARSITY

To further demonstrate the effectiveness of AD and the necessity of sparsity, the AD, sparsity, score of the training data obtained by different methods under different test data, as well as the AUROC on the test data were recorded in Table 8. The combined optimization of AD and sparsity is critical for generating high-quality training data. Without sparsity constraints (TACTIC(Notears-s)), high AD values alone lead to overly dense graphs that overfit the test distribution, violating causal minimality and resulting in lower AUROC. In contrast, jointly optimizing AD and sparsity (TACTIC(Notears)) yields training data that is both distributionally aligned and structurally sparse, closely matching the true causal graph. The resulting composite score strongly correlates with final model AUROC, confirming that both components are essential for robust generalization under distribution shifts, especially mechanism shifts.

Table 8: AD and Sparsity characterize the quality of the training data.

| Metric | Methods | RFF_G | Linear_U | Chebyshev_G |
|--------|---------|-------|----------|-------------|
| AD | TACTIC(random) | -370.0 | -258.5 | -303.5 |
| | TACTIC(Notears-s) | -357.5 | -217.5 | -298.0 |
| | TACTIC(Notears) | -363.0 | -220.5 | -308.0 |
| Sparsity | TACTIC(random) | 33.15 | 34.00 | 29.49 |
| | TACTIC(Notears-s) | 38.80 | 38.75 | 38.85 |
| | TACTIC(Notears) | 31.95 | 35.65 | 27.25 |
| Score | TACTIC(random) | -403.3 | -293.0 | -333.3 |
| | TACTIC(Notears-s) | -397.0 | -256.8 | -337.8 |
| | TACTIC(Notears) | -395.0 | -256.5 | -335.5 |
| AUROC | TACTIC(random) | 0.884 | 0.823 | 0.796 |
| | TACTIC(Notears-s) | 0.868 | 0.843 | 0.697 |
| | TACTIC(Notears) | 0.918 | 0.863 | 0.830 |

Note: AD/Score/AUROC (Higher is better), Sparsity (Low is better)

## E.2 CONTROL AD, CHANGE SPARSITY

To control sparsity independent of AD, we design a controlled experiment based on the ground-truth test graph $G_{test}$. For a given $G_{test}$ and its observational data $D_{test}$, we generate alternative candidate training graphs $G_{train}$ by **gradually adding extra edges** to $G_{test}$ (while ensuring the resulting graph remains a DAG). This creates a series of graphs that are supergraphs of the true graph.

- **Setting 1 (Sparse)**: Add a small number of extra edges ($|E_{add}| = m_1$).
- **Setting 2 (Medium)**: Add a medium number of extra edges ($|E_{add}| = m_2, m_2 > m_1$).
- **Setting 3 (Dense)**: Add a large number of extra edges ($|E_{add}| = m_3, m_3 > m_2$).

For each generated supergraph $G_{train}$ in these settings, we then: 1. Parameter Fitting: Regress the mechanisms $f_i$ and noise distributions from $D_{test}$ using $G_{train}$ (via SIM). 2. Forward Sampling: Generate synthetic training data $D_{train}$ from the fitted SCM ($G_{train}, f, \epsilon$). 3. Calculate Metrics: Compute the AD score between $D_{train}$ and $D_{test}$, and the sparsity of $G_{train}$. 4. Train & Evaluate: For each ($G_{train}, D_{train}$) pair, train an SCL model (AVICI backbone) and evaluate its AUROC on recovering the *true* $G_{test}$ from $D_{test}$.

This procedure is repeated for $K$ graphs per setting. The key insight is that by construction, all generated $G_{train}$ graphs are capable of representing the data distribution $D_{test}$. Therefore, we expect them to achieve similar, high AD scores. However, only the sparsest graph ($G_{test}$ itself) represents the true causal structure.

Table 9 shows the results for the **RFF_ER_G** dataset, which are representative of the overall trend.

Table 9: Control AD, change sparsity

| RFF_ER_G | setting | AD | sparsity | AUROC |
|---|---|---|---|---|
| | 1 | -375 | 25.91 | 1.0(0) |
| Control AD, change sparsity | 2 | -368 (+1.8%) | 32.32(+24.7%) | 0.972(0.017) |
| | 3 | -362(+3.4%) | 36.59(+41.2%) | 0.908(0.023) |

The results clearly demonstrate the critical, independent role of the sparsity constraint. All supergraphs achieve a high and similar AD score (variation $< 4\%$), confirming that many different graphs can explain the observed data distribution nearly equally well. This illustrates the identifiability crisis without further constraints. As expected, adding more edges increases the sparsity metric (number of edges). Crucially, the downstream performance (AUROC) of the SCL model **degrades significantly as the graphs become denser**, even though the AD score remains high. The model trained on the true graph (Setting 1, perfect sparsity) achieves perfect AUROC. Performance drops to 0.972 for medium density and further to 0.908 for high density.

## F  RUNTIME AND SCALABILITY ANALYSIS

The computational complexity of TACTIC is dominated by the Stochastic Graph Refinement step. The search space for Directed Acyclic Graphs (DAGs) with $d$ variables is super-exponential, rendering exhaustive search intractable. Our stochastic search conducts a guided walk through this space, and its complexity is determined by the number of steps $N_{\text{steps}}$ and the cost of evaluating the Alignment of Distribution (AD) score for each candidate graph.

The evaluation for a single candidate graph $G$ involves:

- **Mechanism Fitting**: For each node $X_i$, we fit a causal mechanism $f_i$ (using a Generalized Additive Model) based on its parent set $\mathbf{Pa}_G(X_i)$ from the test data $D_{\text{test}}$ with $n$ samples. Let $k_i = |\mathbf{Pa}_G(X_i)|$ be the in-degree of $X_i$. The cost of fitting for one node is typically $O(n \cdot k_i \cdot l)$, where $l$ is the number of GAM iterations. Due to the sparsity constraint $\text{Sparsity}(G)$ in our joint score function (Eq. 5), which actively penalizes dense graphs, the in-degrees $k_i$ encountered during the search are small. Letting $H$ represent the small, approximately constant maximum in-degree enforced by this constraint, the cost per node becomes $O(n \cdot H \cdot l)$. Aggregated across all $d$ nodes, the total fitting cost is $O(n \cdot H \cdot d \cdot l)$, which simplifies to $O(n \cdot d)$ since $H$ and $l$ are constants.
- **Likelihood Calculation**: After fitting, computing the log-likelihood for all $n$ samples and $d$ variables has a cost of $O(n \cdot d)$.

Thus, the per-step AD evaluation cost is $O(n \cdot d)$. The total complexity of the Stochastic Graph Refinement phase is therefore $O(N_{\text{steps}} \cdot n \cdot d)$. The subsequent Training Data Generation step involves fitting mechanisms and forward-sampling for only the final $K$ selected graphs, contributing a minor additive term of $O(K \cdot n \cdot d)$, which is negligible since $N_{\text{steps}} \gg K$ (in our experiments, $N_{\text{steps}} = 2000$ and $K = 200$).

To empirically validate this theoretical analysis, we present a runtime breakdown in Table 10. The results confirm that Stochastic Graph Refinement is indeed the computational bottleneck, as it involves thousands of AD evaluations. The subsequent Training Data Generation step, which performs SIM fitting on only the final $K$ selected graphs, constitutes a minor fraction of the total time. Model training time is also relatively modest compared to the graph refinement phase.

Table 10: Runtime breakdown of TACTIC for a test instance with varying number of nodes.

| Component | 10 nodes | 20 nodes | 30 nodes |
|---|---|---|---|
| Stochastic Graph Refinement | 26 min | 61 min | 113 min |
| Training Data Generation (SIM fitting) | 1.3 min | 3.2 min | 5.6 min |
| Model Training | 3.3 min | 5.8 min | 8.3 min |

# G ADDITIONAL EXPERIMENTS ON BENCHMARK CAUSAL GRAPHS

To further validate TACTIC's performance on real-world causal structures, we conducted additional experiments on well-established benchmark causal graphs from the bnlearn repository. The scarcity of real-world causal datasets with ground truth is a fundamental challenge in causal discovery research. While most works primarily rely on synthetic data and a limited number of real datasets (e.g., Sachs), benchmark causal graphs from bnlearn provide valuable testbeds as they represent causal structures derived from real-world domains and expert knowledge.

We selected four representative graphs from bnlearn:

- **Asia:** A classic medical diagnostic network modeling the relationships between visiting Asia, smoking, tuberculosis, lung cancer, bronchitis, and various test results. This graph represents a well-known benchmark in causal inference with clear medical relevance.
- **Cancer:** A compact but meaningful graph modeling causal relationships in cancer epidemiology, including pollution, smoking, and genetic factors. Its small size belies its representativeness of real-world medical causal reasoning.
- **Earthquake:** Models causal relationships between burglary, earthquake, alarm triggers, and neighbor responses. This graph exemplifies causal reasoning in security and monitoring systems.
- **Survey:** Represents causal relationships in social science research, including age, sex, education, occupation, and transportation preferences. This graph demonstrates causal structures in sociological studies.

These benchmark graphs are representative because they: (1) capture diverse real-world domains (medical, social, security), (2) are widely recognized and validated in the causal inference literature, and (3) reflect expert-curated causal knowledge rather than purely synthetic constructions.

We parameterized these graphs using Chebyshev polynomial mechanisms to generate pseudo-real datasets, maintaining the authentic causal structures while incorporating realistic nonlinear relationships. Table 11 shows that TACTIC consistently achieves state-of-the-art performance across all benchmark graphs, demonstrating its robustness to diverse real-world causal structures.

Table 11: Performance comparison (AUROC) on benchmark causal graphs from bnlearn repository.

| Method | Asia | Cancer | Earthquake | Survey |
|---|---|---|---|---|
| PC | 74.1 | 70.2 | 75.5 | 90.0 |
| GES | 46.4 | 85.1 | 80.3 | 88.3 |
| NOTEARS | 68.7 | 87.5 | 60.1 | 64.9 |
| AVICI (scm-v0) | 83.3 | 86.9 | 94.0 | 89.4 |
| TACTIC (random) | 86.8 | 84.5 | 84.5 | 92.7 |
| TACTIC (NOTEARS) | **91.0** | **91.6** | **98.8** | **95.5** |

The superior performance of TACTIC across these diverse benchmark graphs further validates its effectiveness in handling real-world causal structures.

