# OpenReview forum: "Test Time Training for Supervised Causal Learning"
_ICLR.cc/2026/Conference — Submitted to ICLR 2026_

### Official Review · Reviewer_mjjp · 2025-10-16

**Soundness:** 2
**Presentation:** 2
**Contribution:** 3
**Rating:** 4
**Confidence:** 4

**Summary:**

This paper introduces TTT-SCL (Test-Time Training for Supervised Causal Learning), to address the poor out-of-distribution generalization of current supervised causal discovery methods.

The authors first show through experiments that current supervised methods suffer from fragility to distribution shifts, failure in compositional generalization, and a significant performance gap between synthetic benchmarks and real-world data.  They highlight the need of "diversity" in training data.

The authors then propose to generate training data at test time that is aligned closer to test instances.  The underlying causal model is assumed to be additive noise models.  Under this, the candidate causal graph is regressed against the test data, and ones with high regression consistency are admitted into training set, together with their forward sampled data.

Experiments show that TTT-SCL achieves stronger and more consistent performance than other pre-trained supervised methods.

**Strengths:**

1. The goal is well-motivated and timely.  The authors correctly identify a real gap in the supervised causal discovery literature: the strong in-distribution performance but poor robustness under realistic domain shifts.

2. Their diagnosis through controlled experiments (mechanism/graph/noise shifts) is thorough and convincing to reveal the above issues.

3. The whole method procedure and the graph refinement strategy are simple and heuristic.  They seem to perform well though, via the experimental results reported by authors.

**Weaknesses:**

1. **The work lacks any kind of formal theoretical analysis:**
- While it is understandable that their identifiability results are rooted back to the additive noise models setting, the key arguments in this paper itself still need more justification.
- For instance, at line 281, ".. if.. closely aligned.. (Dk≈Dtest), then .. is likely a close approximation.. (Gk≈Gtest)."  The authors need to discuss the scale of "≈":  Do same levels graph differences (e.g., measured by structural hamming distance) always lead to the same levels of differences in the sampled data?  Should this "alignment of distribution" score, as well as the sparsity constraint, be normalized by the sample size of data or graph size?  These problems need to be addressed to ensure the supervised method is grounded.
- Moreover, within the scope of this work the unique identification of the DAG is guaranteed by ANM.  Then what about other cases where only an equivalence class is identifiable?  How to characterize the "Gk≈Gtest" in that case?  These should be discussed.

2. **Search space for training instances can be improved to prevent from overfitting:**  Currently only the new graphs are generated for test-time training, together with the data forward sampled from them using the parameters regressed on data.  However, these regressed parameters can be arbitrarily complex, leading to overfitting issues.  Could the authors please explain why new data generating functions are not generated together with graphs, and in a way similar to regularize the sparsity for graphs, regularize the complexity of these functions?

3. **Search strategy for training instances can also be improved for efficiency:**  Currently the "optimal new training graphs" are searched through stochastic graph refinement.  It seems that many existing optimization based causal discovery methods can be incorporated as a component to directly optimize the AD score.  Existing unsupervised causal discovery methods may also be used to get some "base graphs" to modify from.  Following this, some existing supervised causal discovery methods may also be viewed as "test-time training".

4. **The overall presentation and writing is poor:**  For instance,
- The authors introduce unnecessary technical notations even in the start of the first introductory paragraph.
- The whole section 3.1 is hard to read, where more natural language are expected to explain the experimental setup.
- The term "low sparsity" in Figure 2 can be ambiguous.
- Too much abbreviation abuse throughout the paper.  What is the authors' proposed method, TTT-SCL or TACTIC?  The abbreviation for AD seems unnecessary, and subscripts with ("G", "U") noise notations only make section 3.1 hard to parse.  At the same time, many abbreviations are used before explaining them, e.g., the "DAG" is used at line 37 but only declared at line 104.

**Questions:**

see "weaknesses"

---

> ### Author Response · Authors · 2025-11-24
> **Response to Reviewer mjjp (1/3)**
>
> We appreciate the reviewer's efforts invested in reviewing our work. We have carefully considered each of your detailed comments and have addressed them one by one to alleviate your concerns.
>
> > **0. Overall Response to Reviewer mjjp**
>
> We emphasize that this work is a preliminary exploration aimed at filling the unexplored area of how SCL should address generalization. By identifying the fundamental limitations in current SCL practice, we demonstrate the necessity of a paradigm shift from static pre-training to instance-aligned training. By introducing the idea of test-time training into SCL, TTT-SCL can be viewed as a foundational framework—demonstrating the viability of this shift and opening the path for future work to develop stronger theoretical guarantees, more expressive mechanism models, and more efficient graph-search procedures.
>
> > **Reviewer:** *1.1 For instance, at line 281, "... if.. closely aligned.. (Dk$\approx$Dtest), then .. is likely a close approximation.. (Gk$\approx$Gtest)." The authors need to discuss the scale of "$\approx$": Do same levels graph differences (e.g., measured by structural hamming distance) always lead to the same levels of differences in the sampled data?*
>
> Our framework relies on the identifiability of ANMs as its foundational premise, ensuring that perfect distribution alignment in the infinite-data limit implies the true graph structure. Regarding the scale of ``$\approx$'', we admit that we currently lack a precise, quantitative theoretical bound linking the AD score difference to the structural distance between $G^k$ and $G_{test}$. This is a challenging and open problem, as the relationship is highly dependent on the specific functional forms and noise distributions. Our current work provides strong empirical evidence that our proposed score function is effective in practice, but we agree that a formal characterization of this approximation is an important goal for future theoretical research.
>
> > **Reviewer:** *1.2 Should this "alignment of distribution" score, as well as the sparsity constraint, be normalized by the sample size of data or graph size?*
>
> AD Term: It is normalized by the number of variables $d$ (as in Eq.~3), making scores comparable across graphs of different sizes. The log-likelihood in AD scales linearly with $n$. Since we compare scores for a fixed test instance (fixed $n$), this acts as a constant factor and does not affect the relative ranking of candidate graphs.
>
> Sparsity Term: It is defined as the L$_0$ norm (number of edges $|E|$), whose scale depends on graph size $d$. We acknowledge that from a pure optimization perspective, having both terms be scale-invariant is ideal. We employed current formulation for two key reasons, which were validated empirically:
>
> - Implicit Balancing via $\lambda$: The hyperparameter $\lambda$ is precisely meant to balance two terms of different natures and scales. The fact that a single $\lambda=1$ worked robustly across our experiments with $d \in \{10, 11, 20\}$ demonstrates that this balance is stable within the scope of our study.
> - Theoretical Prior from AIC: Our choice was motivated by the Akaike Information Criterion (AIC), where the complexity penalty is the unnormalized number of parameters $k$. Our formulation, $\text{score} = \text{AD} - \lambda\cdot|E|$, directly mirrors the AIC structure $-2\ln(L) + 2k$, providing a principled prior for $\lambda=1$.
>
> > **Reviewer:** *1.3 Moreover, within the scope of this work the unique identification of the DAG is guaranteed by ANM. Then what about other cases where only an equivalence class is identifiable? How to characterize the "Gk$\approx$Gtest" in that case?*
>
> In cases where only an MEC is identifiable, the interpretation of $G^k \approx G_{test}$ should indeed be that they belong to the same equivalence class. Our framework can be naturally extended to this setting. The AD metric remains valid because all graphs in the same MEC imply the same joint distribution and would thus achieve the same ideal AD score. The sparsity constraint would then select the sparsest member(s) within that class. We will add a discussion on this point and the potential for learning equivalence classes as a promising future extension.

---

> ### Author Response · Authors · 2025-11-24
> **Response to Reviewer mjjp (2/3)**
>
> > **Reviewer:** *2. Search space for training instances can be improved to prevent from overfitting.*
>
> The potential for overfitting is indeed a critical consideration when the mechanism parameters are regressed from the test data for each candidate graph.
>
> We deliberately chose this design for a critical reason: to ensure the generated training data is both causally meaningful and distributionally aligned. The core idea of TTT-SCL is to create a training set $\{(D^k_{train}, G^k_{train})\}$ where the data $D^k_{train}$ is a faithful realization of the causal structure $G^k_{train}$. Using the Structure-Induced Mechanism (SIM) step is the most direct way to enforce this faithfulness. For a given $G^k_{train}$, regressing its specific mechanisms from $D_{test}$ ensures that the forward-sampled $D^k_{train}$ comes from a model that is semantically consistent with $G^k_{train}$ and, if $G^k_{train}$ is a good candidate, distributionally close to $D_{test}$.
>
> To prevent the overfitting that the reviewer rightly points out, our framework incorporates two key, synergistic regularization components:
>
> - Sparsity Constraint on Graph Structure: This is our primary tool to counteract the selection of overly complex graphs that might achieve a high AD score by introducing spurious edges. It enforces the principle of causal minimality, directly regularizing the complexity of the causal hypotheses we generate.
> - Inherent Regularization in Mechanism Fitting: Crucially, we avoid using highly flexible function approximators like neural networks for the SIM step, which could easily overfit. Instead, we use relatively simple and constrained function classes, specifically the **Generalized Additive Model (GAM)**, for mechanism regression. This choice inherently limits the complexity of the fitted functions $f_i$, preventing them from memorizing noise in the test instance and thus acting as a form of implicit regularization.
>
> We agree with the reviewer that jointly regularizing both the graph complexity and the functional complexity of the mechanisms is a profound and promising direction. Exploring this, for instance, by explicitly penalizing the complexity of the fitted functions $f_i$, could further robustify the framework. We will consider this a valuable avenue for future research.

---

> ### Author Response · Authors · 2025-11-24
> **Response to Reviewer mjjp (3/3)**
>
> > **Reviewer:** *3. Search strategy for training instances can also be improved for efficiency.*
>
> Our current stochastic graph refinement is a foundational proof-of-concept. The reviewer's suggestion to incorporate graphs from unsupervised causal discovery methods as "base graphs" aligns perfectly with the flexible, plug-and-play philosophy of the TTT-SCL framework. In fact, our TACTIC (Notears) variant, which uses a NOTEARS-estimated graph as the seed, is a direct implementation of this idea and shows clear performance gains over a random seed (see Table~3). We will emphasize this flexibility and the value of good initialization more clearly in the revision.
>
> We argue that there is a fundamental, qualitative difference between the amortized inference of standard SCL and the dynamic instance-specific training of TTT-SCL.
>
> - Standard SCL (Amortized Inference): A model is pre-trained once on a fixed, static distribution of training graphs. At test time, it performs a forward pass. The model's parameters and the knowledge embedded within them are **fixed**. It cannot adapt its fundamental understanding of causality based on the specific test instance; it can only query its pre-existing knowledge.
> - TTT-SCL (Dynamic Training): For each test instance, we dynamically generate a **new, customized** training set explicitly aligned with that instance. An SCL model is then trained from scratch on this set. The model's parameters are specifically optimized for this particular test instance. The "knowledge" of the model is created on-the-fly and is unique to the test instance.
>
> The critical distinction is that standard SCL uses a fixed function to map data to graphs, while TTT-SCL dynamically defines the learning problem itself (the training data) for each test instance. This allows TTT-SCL to break free from the constraints of the pre-training distribution, which is the root cause of the generalization failures we identified. Therefore, we posit that viewing standard SCL as test-time training would be a mischaracterization of its fixed, amortized nature. Our TTT-SCL framework represents a genuine paradigm shift from static, one-size-fits-all pre-training to dynamic, instance-specific adaptation.
>
> >**Reviewer:** *4. The overall presentation and writing is poor*
>
> We sincerely thank the reviewer for this feedback regarding the presentation and writing of our manuscript. We are now undertaking a comprehensive revision to address these issues. Specifically, we will undertake the following actions to significantly improve the manuscript:
>
> - Thorough Polishing: We will perform a line-by-line edit of the entire paper to eliminate unnecessary technical notations, reduce abbreviation abuse, define all terms upon their first appearance, and improve the overall flow and readability of the text.
> - Restructuring of Section 3: We will rewrite Section 3 using clear, natural language and logical progression to make the experimental design easy to follow.
> - Clarification of Terminology: We will unify and clarify key terms throughout the paper. For instance, we will replace ambiguous labels like **Low Sparsity** in figures with clearer descriptions like **Sparse Graph**, and carefully explain all data generation symbols.
> - Methodological Distinction: We will ensure a consistent and clear distinction between our proposed framework (**TTT-SCL**) and its first concrete instantiation (**TACTIC**).
> - Improved Visual Communication: We will redesign the presentation of key results (e.g., in Table~1) to use more intuitive visual summaries where appropriate, as suggested by another reviewer, to deliver the message more effectively.
>
> A revised version of the manuscript incorporating all these improvements, along with the other changes detailed in our rebuttal, will be uploaded **before the rebuttal period ends**. We are confident that these efforts will transform the manuscript into a much more clear, professional, and impactful piece of work.

---

> ### Comment · Reviewer_mjjp · 2025-11-27
>
> Thank the authors for the response. My concerns about the search space (related to overfitting) and search strategy (related to efficiency) are partly addressed, in that this work is mainly a first proof of concept to show the necessity of test time training for supervised causal discovery. I have raised my score accordingly.
>
> As for my concern on the theoretical analysis (e.g., those related to "≈"), I hope the authors could also incorporate these clarifications, as well as the writing issues, into the update.

---

### Official Review · Reviewer_Dm4H · 2025-10-18

**Soundness:** 2
**Presentation:** 3
**Contribution:** 3
**Rating:** 4
**Confidence:** 4

**Summary:**

This work firstly shows that traditional methods for Supervised Causal Learning (SCL) based on pretrained models that map a dataset $D$ to a graph $G$ fail to generalize to out-of-distribution data.  To address this issue, the Test Time Training for Supervised Causal Learning (TTT-SCL) framework is introduced - along with an algorithm, TACTIC, that implements it. In a nutshell, TACTIC generates a set of graph candidates that maximize a sparsity-regularized goodness-of-fit score for the test data. Then, it produces datasets $\{D_{t}\}_{t}$ for each graph by regressing a predefined causal mechanism into the test data. Finally, it fine-tunes an existing SCL algorithm (AVICI) on the sampled graph-data.

Overall, the introduced approach is seemingly sound, clearly positioned in the realm of SCL algorithms, simple, and demonstrably effective through well-executed experiments. However, I have a few concerns regarding exposition and reproducibility (listed below), and I will be happy to give a positive review for the work in case they are addressed.

**Strengths:**

1. Both TTT-SCL and TACTIC are clearly described and properly evaluated.

2. Limitations of prior works are nicely discussed and experimentally assessed.

3. The method has clear advantages over existing, test-data-independent, approaches, which are confirmed by comprehensive experiments.

4. Figures 1 and 2 are quite helpful in understanding the TACTIC.

**Weaknesses:**

1. Code has not been publicly released, which is a major drawback for reproducibility  - especially given the empirical nature of the work. Will the authors release the code?

2. As I understand, the generated graph-data pairs are either used for training a SCL from scratch or fine-tuning an existing SCL, although I am not sure which approach is undertaken. Could the authors clarify this? I believe this is a major component for the work that lacks a clear description.

3. If it is only training from scratch, the optimization problem could be formulated as

$$
	\min\_{G} \mathbb{E}\_{G \sim P\_{test}} [ L(G, D\_{test}) | D\_{test}], % . % .
$$

for whichever (empirical) distribution $P\_{test}$ is induced by the score function in Equation (5) and a loss function $L$ that defines the training of the underlying SCL model (AVICI). Is this correct (Otherwise, how can we compare fine-tuning vs. training the SCL model from scratch?) In this case, TACTIC is equivalent to score-based Causal Discovery algorithms - with a specific, stochastic score - and I believe that the distinction in Figure 1 is inaccurate.

4. Out of the three presented issues (lack of robustness to distribution shift, failure to perform compositional generalization, and mismatch between real and synthetic tasks), only two of them have been assessed on the experiments, as far as I understand. Could the authors elaborate on how TACTIC addresses compositional generalization (perhaps through experiments)?

**Questions:**

Please refer to the section above.

---

> ### Author Response · Authors · 2025-11-24
> **Response to Reviewer Dm4H (1/2)**
>
> We would like to express our gratitude for your constructive and valuable feedback. Your question helped us to better clarify and improve our paper. Please see our point-by-point response below.
>
> > **Reviewer:** *1. Will the authors release the code?*
>
> Yes, we have released the code through the anonymous platform; the link is https://anonymous.4open.science/r/TACTIC-B1B0/
>
> > **Reviewer:** *2. As I understand, the generated graph-data pairs are either used for training a SCL from scratch or fine-tuning an existing SCL, although I am not sure which approach is undertaken. Could the authors clarify this?*
>
> In our current implementation, TACTIC trains the underlying SCL model **from scratch** on the dynamically generated graph-data pairs for each test instance. This design choice was made to provide the most direct validation of our core thesis: the primary bottleneck for robust generalization in SCL is the misalignment between the static pre-training distribution and the test instance, and this can be solved by generating a minimal, aligned training set at test time.
>
> > **Reviewer:** *3. If it is only training from scratch, TACTIC is equivalent to score-based Causal Discovery algorithms - with a specific, stochastic score - and I believe that the distinction in Figure 1 is inaccurate.*
>
> We thank the reviewer for this comment, which allows us to clarify a key conceptual aspect of our framework. We wish to emphasize a fundamental distinction in **objective and outcome** between TACTIC (the implementation of TTT-SCL framework) and classical score-based causal discovery:
>
> - **Classical Score-based Methods**: The primary goal is to **directly output the high-scoring graph** found during the optimization process. The score function is the final objective.
> - **TACTIC / TTT-SCL**: The primary goal is **not** to output the high-scoring graph itself. Instead, the goal is to use the score to **generate a high-quality, aligned training set** $\{(D^k_{train}, G^k_{train})\}_{k=1}^K$. The final causal graph is predicted by an SCL model that is **trained from scratch on this generated set**. We optimize the quality of the training data to activate the powerful inductive bias and generalization capability of the SCL model, rather than using the score to directly select a graph.
>
> Therefore, while the search mechanism shares similarities with score-based methods, TTT-SCL leverages graph search as an internal engine for data generation, with the final output being determined by a **trained SCL model**. This is the core difference illustrated in Figure~1.
>
> Based on this clarification, we agree that the presentation in Figure~1 could be more precise. We will revise it in the new version, changing **Traditional CD** to the more general **Non-SCL CD Methods** and more clearly depicting that TTT-SCL's final output comes from a dynamically trained SCL model.

---

> ### Author Response · Authors · 2025-11-24
> **Response to Reviewer Dm4H (2/2)**
>
> > **Reviewer:**  *4. Out of the three presented issues (lack of robustness to distribution shift, failure to perform compositional generalization, and mismatch between real and synthetic tasks), only two of them have been assessed on the experiments, as far as I understand. Could the authors elaborate on how TACTIC addresses compositional generalization (perhaps through experiments)?*
>
> The evaluation of compositional generalization (Issue 2) is contained in our manuscript, specifically in **Section 3.2 and Table 1, through the `AVICI (mixed)`** experimental setting.
>
> To be precise, the `AVICI (mixed)` model was trained on a mixture of data from the following six distributions: RFF\_U\_ER, RFF\_U\_SF, Linear\_G\_ER, Linear\_G\_SF, Chebyshev\_U\_ER, and Chebyshev\_U\_SF. Crucially, this training set contains all individual component types (RFF/Linear/Chebyshev mechanisms, ER/SF graphs, Gaussian/Uniform noise) but **systematically omits specific combinations** used for testing, such as RFF\_G\_ER and Linear\_U\_SF.
>
> We designed **Response Table 6** to present this result to you more clearly. The performance drop of `AVICI (mixed)` compared to the `i.i.d.` setting is the direct evidence of the compositional generalization failure. The model fails to perform well on test combinations it has not seen, even though it has seen all the constituent elements separately.
>
> **Response Table 6: Evaluating combinatorial generalization.**
> |                          | **RFF_G_ER** | **RFF_G_SF** | **Linear_U_ER** | **Linear_U_SF** | **Chebysev_G_ER** | **Chebysev_G_SF** |
> | ------------------------ | ------------ | ------------ | --------------- | --------------- | ----------------- | ----------------- |
> | iid                      | 90.0 (2.7)   | 100.0 (0.0)  | 91.7 (4.3)      | 100.0 (0.0)     | 93.0 (2.9)        | 100.0 (0.0)       |
> | AVICI (mixed)         | 84.8 (4.7)   | 89.0 (2.0)   | 88.0 (3.8)      | 87.9 (5.9)      | 82.6 (6.5)        | 89.0 (9.5)        |
>
> The core of Issue 2 is a combinatorial generalization problem, which is distinct from simply adding more types of distributions to the training set. This exposes a fundamental limitation of static pre-training: covering the entire combinatorial space of possible instances is intractable. In contrast, TACTIC fundamentally bypasses this problem by dynamically generating a training set explicitly aligned with the specific test instance $D_{test}$. It does not need to in advance cover all possible combinations but creates the most relevant training distribution on-demand for each test instance.

---

> > ### Comment · Area_Chair_DCti · 2025-11-28
> >
> > Dear Reviewer,
> >
> > Please make sure you read the authors' response and engage with them in the discussion before the end of the discussion period on **Dec 03 '25 09:00 PM UTC**. This is a hard deadline.
> >
> > Thank you for supporting quality peer review at ICLR.
> >
> > AC

---

> > ### Comment · Area_Chair_DCti · 2025-11-28
> >
> > Dear Reviewer,
> >
> > Please make sure you read the authors' response and engage with them in the discussion before the end of the discussion period on **Dec 03 '25 09:00 PM UTC**. This is a hard deadline.
> >
> > Thank you for supporting quality peer review at ICLR.
> >
> > AC

---

> ### Author Response · Authors · 2025-11-28
>
> We appreciate the reviewers' efforts. **And we kindly remind you that we have released our code and provided usage guidelines to ensure the reproducibility of our work.** The link is https://anonymous.4open.science/r/TACTIC-B1B0/.
>
>
> To further address your question 3, we provide concrete evidence demonstrating that TTT-SCL represents a fundamentally different approach from score-based causal discovery methods. Specifically, we conducted additional experiments comparing three different outputs using one test instance across four test domains (RFF, Linear, Chebyshev, and Sachs) for detailed analysis:
>
> 1. **Seed graph**: Initial graph from proxy methods
> 2. **Highest-score graph**: Best graph found during TACTIC's stochastic refinement
> 3. **Final output**: Graph predicted by the SCL model trained on TACTIC-generated data
>
> The results in **Response Table 7** clearly demonstrate the two-stage improvement of our approach:
>
> - **1->2 (Search Improvement)**: The higher AUROC of the highest-score graph compared to the seed graph shows that TACTIC's stochastic refinement effectively improves graph quality through distributional alignment.
>
> - **2->3 (Learning Improvement)**: The consistent and substantial performance gain of the final SCL output over the highest-score graph demonstrates the crucial advantage of our approach. While score-based methods would stop at the highest-scoring graph, TACTIC uses this graph to generate training data that enables an SCL model to learn more accurate causal relationships.
>
> This two-stage process, where we optimize for training data quality rather than directly for the final graph, constitutes the fundamental distinction between TACTIC and classical score-based causal discovery.
>
> **Response Table 7**: Performance comparison (AUROC) across different stages of TACTIC
>
> | | RFF | Linear | Chebyshev | Sachs |
> |--------------|-----|--------|-----------|-------|
> | Seed graph | 0.784 | 0.748 | 0.529 | 0.618 |
> | Highest-score graph in TACTIC search | 0.895 | 0.770 | 0.694 | 0.666 |
> | Final graph from trained SCL model | **0.918** | **0.894** | **0.800** | **0.789** |

---

### Official Review · Reviewer_V2uH · 2025-10-21

**Soundness:** 2
**Presentation:** 2
**Contribution:** 1
**Rating:** 2
**Confidence:** 5

**Summary:**

The authors illustrate limitations of supervised causal discovery, and propose a framework (TTT-SCL) and a method (TACTIC) to overcome these limitations. In particular, given that the main issue is the inability of SCL methods to generalize out of distribution, they propose to generate data at test time for fine tuning/training of an SCL method.

**Strengths:**

The studied problem is important and timely. An interesting contribution is the empirical evidence of failure of AVICI to generalize to Sachs nd Syntren, two (pseudo) real datasets.

**Weaknesses:**

1. **Novelty.** The first two issues identified in the paper about SCL are known e.g. from Montagna et al., 2024, which the authors fail to mention and discuss in relation to their work. The third issue is nice to be observed, but it is a corollary of the other two: if we show that SCL fails OOD  in synthetic settings, the same should happen in real settings (but it is nice to see! Not sure somebody showed it before)
2. **Novelty.** In relation to Issue 2: the same problem is observed in Montagna et al., 2024 (figure 4 in their paper). However, they show that the performance gap is closed as you increase the number of training datasets (see figure 12 in Montagna et al., 2024). In your setting, it is unclear how many datasets from each SCM type (eg RFF_G_ER is one SCM type) are used in the mixed setting, so it’s also unclear whether the conclusions in Montagna et al., 2024 would apply here. This requires discussion.
3. Given that the novelty in the identified issues in SCL is at best only partial, the main contribution is the algorithm to mitigate distribution shifts. I see two big issues here
    1. **Statistical significance of experiments**. Conclusions of improved performance of TACTIC on (pseudo) real data can not be drawn only from two datasets (Sachs and Syntren)
    2. **Methodological/conceptual.** TTT-SCL/TACTIC, in order to work, requires training/fine-tuning on the test data **assuming they were generated according to an identifiable SCM;** in the paper, an additive noise model. Then (as already discussed in Montagna et al., 2024), this is equivalent to restricting the space of solutions under the assumptions of an additive noise model: if we are willing to make this assumption, why use SCL in the first place? I can use any of the known methods for causal discovery on additive noise models (SCORE, RESIT, NOGAM, GranDAG, …). The clear advantage of these methods is that they do not require a training procedure, so there is not even the problem of distribution shifts: e.g., there is no issue of noise distribution shift, as most of these methods simply don’t need assumptions on the noise distribution. In this light, I don’t see the improvement that TTT-SCL/TACTIC brings to the table of algorithmic causal discovery.
4. Other issues
    1. **Reproducibility.** The authors fail to provide sufficient specifications on the training
        1. In the TACTIC procedure, how do you choose the noise distribution of the SCM
        2. In TACTIC, how do you choose the algorithm to sample the graph?
        3. In section 3.2, how many datasets do you use for training, each one with how many samples?
        4. In section 3.2, what are the specifics for AVICI training
    2. I am not sure NOTEARS, GES, and PC are the most probing methods to compare to. Since this is mostly an algorithmic paper, many more comparisons are needed (SCORE, CAM, NoGAM, RESIT provide good baselines for ANM causal discovery — these are just some of my picks, feel free to change/expand)
    3. **Clarity**.
        1. In section 3.2, the choice of presenting the results with a table, no highlights, doesn’t really help to deliver the message. Please consider using figures?
        2. L212: “Each specific training data setting is indicated above the result”. I assume this refers to tables? Please clarify this, and also consider making this explicit in the tables’ captions.
    4. L222: “AVICI can be considered the strongest model of open source under the SCL paradigm”: any evidence for this?

**Questions:**

Please refer to the weaknesses section

---

> ### Author Response · Authors · 2025-11-24
> **Response to Reviewer V2uH (1/4)**
>
> We appreciate the reviewer's efforts invested in reviewing our work. We have carefully considered each of your detailed comments and have addressed them one by one to alleviate your concerns.
>
> >**Reviewer:** *1. The reviewer claims that our issues regarding SCL generalization have been identified.*
>
> We thank the reviewers for pointing out the work by Montagna et al., 2024, which we are pleased to see shares a similar concern regarding the generalization of SCL. This convergence of independent research further underscores the critical importance of this problem.
>
> However, we **disagree** with the assessment that our issues are subsumed by prior observations. While investigating generalization in supervised paradigms is indeed natural, what matters more are the specific methodology for diagnosing SCL, the conclusions drawn, and the solution pathway these enable. Our systematic diagnosis of SCL's failure modes, particularly the identification of the **combinatorial generalization** problem and the experimentally confirmed **synthetic-real gap**, constitutes the central motivation and the **primary contribution** of our work. This profound limitation, which directly motivates our TTT-SCL paradigm, is **distinct** from the discussions in Montagna et al., 2024.
>
> The most fundamental difference: Montagna et al., 2024 focus on the benefit of adding more types of distributions (e.g., training on $A+B$ is better than $B$ for testing on $A$ ), advocating for increased diversity and size in pre-training (which is called "mixture"). In contrast, our Issue 2 exposes a more fundamental **combinatorial explosion** problem, which cannot be resolved by simply increasing the number of mechanism types or training samples.
>
> -   Suppose the space of causal instances is defined by the Cartesian product of possible mechanisms $\{A, B, C\}$, graphs $\{X, Y, Z\}$, and noise types $\{U, V, W\}$. To perform well on a test instance$ A$\_$X$\_$U$, a model must have been trained on instances from the exact combination $ A$\_$X$\_$U$, not merely on the constituent components $A$, $X$, and $U$ separately.
> -   This creates an intractable requirement for static pre-training: covering $3 \times 3 \times 3 = 27$ combinations versus just $3 + 3 + 3 = 9$ components. Our ``AVICI (mixed)'' experiments in Table~1 are designed precisely to demonstrate this specific failure mode, which persists even when all individual components are present in training.
> -   In the second point of the reply, we will further explain this issue through experiments. Even if we increase the sample size, there is no significant improvement in performance.
>
> This combinatorial problem is the **root cause** that makes robust generalization via static pre-training inherently intractable. It is this diagnosis that motivates our paradigm shift from exhaustive pre-training (seeking diversity) to targeted, on-demand data generation (enabling concentration) via TTT-SCL. **We will add the relevant discussion with Montagna et al., 2024 to the Related Works section.**

---

> ### Author Response · Authors · 2025-11-24
> **Response to Reviewer V2uH (2/4)**
>
> >**Reviewer:** *2. The reviewer claims that our issue 2 had already been raised, and wondered whether increasing the sample size would help alleviate the generalization problem.*
>
> As mentioned in the first point of our response, our issue 2 and the finding of Montagna et al., 2024 are two **different** points. To further demonstrate this, we designed the following experiment, along with explanations of the relevant parameters.
>
> We note that in Montagna et al., 2024, the basic training data setup is "Unless otherwise specified, in our experiments we train CSIvA on a sample of 15,000 synthetically generated datasets, consisting of 1,500 i.i.d. observations"; while in Fig. 12 it is "were trained on 15,000 samples for each SCM type (so the "lin,nl" model saw 30,000 samples in total, and the "all" model saw 45,000)". In our setup (Table 1), it is 2,000 datasets and 200 i.i.d. observations. This setup is relatively small, so we conducted further experiments, increasing the number of datasets to 12,000 to observe the performance changes. The results are shown in **Response Table 3**.
>
> **Response Table 3: Evaluating combinatorial generalization: increasing the number of training datasets does not lead to performance improvement.**
> |                          | **RFF_G_ER** | **RFF_G_SF** | **Linear_U_ER** | **Linear_U_SF** | **Chebysev_G_ER** | **Chebysev_G_SF** |
> | ------------------------ | ------------ | ------------ | --------------- | --------------- | ----------------- | ----------------- |
> | iid                      | 90.0 (2.7)   | 100.0 (0.0)  | 91.7 (4.3)      | 100.0 (0.0)     | 93.0 (2.9)        | 100.0 (0.0)       |
> | AVICI (mixed_2k)         | 84.8 (4.7)   | 89.0 (2.0)   | 88.0 (3.8)      | 87.9 (5.9)      | 82.6 (6.5)        | 89.0 (9.5)        |
> | AVICI (mixed_12k)        | 84.2 (3.6)   | 88.5 (3.2)   | 84.7 (4.2)      | 87.9 (3.2)      | 85.0 (5.7)        | 85.7 (8.9)        |
>
>
> The results show that AVICI (mixed_12k) does not show a significant performance improvement over AVICI (mixed_2k), and there is still a certain gap in performance based on iid. This demonstrates that the combinatorial generalization failure is not simply a data scarcity issue but a fundamental limitation of the static pre-training paradigm. This finding powerfully validates our core motivation for shifting to a test-time training paradigm, which dynamically generates the required combined distribution for each specific test instance.

---

> ### Author Response · Authors · 2025-11-24
> **Response to Reviewer V2uH (3/4)**
>
> > **Reviewer:** *3.1. Conclusions of improved performance of TACTIC on (pseudo) real data can not be drawn only from two datasets (Sachs and Syntren)*
>
> Note that the scarcity of real-world datasets is one of the key practical problems in causal discovery, so almost all existing work also does experiments mainly on synthetic datasets and on limited real world datasets (e.g., Ke et al. 2022, Froehlich & Koeppl 2024 and Zhang et al. 2025). Therefore, our settings are consistent with the literature and we believe our presented results on both synthetic data and real data have verified our claims.
>
> Nonetheless, to further verify the improved performance of TACTIC, we selected real-world causal graphs from the bnlearn repository and parameterized (Chebyshev mechanism) them to obtain additional pseudo real datasets. The results are shown in **Response Table 4**. The results show that TACTIC exhibits robust and excellent performance on these pseudo real datasets.
>
> **Response Table 4: Additional pseudo-real experiments on benchmark causal graphs from bnlearn**
> |                  | cancer | earthquake | survey | asia  |
> | ---------------- | ------ | ---------- | ------ | ----- |
> | PC               | 70.2   | 75.5       | 90.0   | 74.1  |
> | GES              | 85.1   | 80.3       | 88.3   | 46.4  |
> | Notears          | 87.5   | 60.1       | 64.9   | 68.7  |
> | AVICI(scm-v0)    | 86.9   | 94.0       | 89.4   | 83.3  |
> | TACTIC(random)   | 84.5   | 84.5       | 92.7   | 86.8  |
> | TACTIC(Notears)  | **91.6** | **98.8**   | **95.5** | **91.0** |
>
> > **Reviewer:** *3.2 The reviewer believes that if TTT-SCL is limited by the ANM assumption, then it does not bring any improvement to the field of causal discovery.*
>
> Because TTT-SCL extracts information from test data, it outperforms traditional causal discovery and static SCL methods regardless of whether the ANM assumption is met.
>
> First, even when the data perfectly conforms to an ANM, TTT-SCL inherits the advantages of SCL and offers **unique strengths over traditional causal discovery methods**, as shown by its superior performance on the Linear_U setting across Table 3 and the expanded baselines in Response Table 5. SCL models, especially those using neural networks (such as Transformers), are powerful function approximators. They can learn to map data patterns to causal structures, thus handling highly complex nonparametric mechanisms. For traditional methods, these mechanisms are difficult to explicitly specify and fit in a single instance, as discussed by Ke et al. (2022) and Zhang et al. (2025). TTT-SCL can leverage the advantages of SCL even on ANM data that has never been seen before.
>
> Second, the strength of the TTT-SCL framework lies in its access to the test instance $D_{test}$, enabling it to achieve **good practical performance even without ANM assumption**. In reality, no real data perfectly conforms to the ANM assumption; for example, the Sachs real dataset is not ANM (Dhir et al. 2024 and Filippi et al. 2016), yet TTT-SCL still achieves state-of-the-art performance on it, as mentioned in Table 3. This is precisely the advantage of our TTT-SCL: on any real data, based on the characteristics of the data, it specifically generates training data and then trains SCL to achieve good performance.

---

> ### Author Response · Authors · 2025-11-24
> **Response to Reviewer V2uH (4/4)**
>
> > **Reviewer:** *4.1.1 In the TACTIC procedure, how do you choose the noise distribution of the SCM*
>
> In the TACTIC procedure, the noise distribution for the SCM during the forward-sampling step (Training Data Generation) is set to a standard Gaussian distribution by default.
>
> > **Reviewer:** *4.1.2 In TACTIC, how do you choose the algorithm to sample the graph?*
>
> The process is as follows as mentioned in 4.2: We start from a seed graph. In each step of the Stochastic Graph Refinement, a new candidate graph is generated by applying a random local modification (edge addition, deletion, or reversal) to the current graph, ensuring the result remains a DAG. The acceptance of this candidate is determined by a stochastic decision based on the score(G) of the candidate and the current graph.
>
> > **Reviewer:** *4.1.3 In section 3.2, how many datasets do you use for training, each one with how many samples?*
>
> In the experiments of Section 3.2, all static training setups (i.i.d., shift, mixed) use a total of 2,000 synthetic training datasets. Each dataset contains n=200 i.i.d. observations. Specifically, for the AVICI (mixed) configuration, these 2,000 datasets are uniformly distributed across the 6 mechanism-graph-noise combinations, resulting in approximately 330 datasets per specific combination.
>
> > **Reviewer:** *4.1.4 In section 3.2, what are the specifics for AVICI training*
>
> For the baseline AVICI (scm-v0), we used the open-source, pre-trained model provided by the authors. We utilized this model directly in its published state without any modification or re-training, ensuring a fair comparison to the officially reported SCL method.
>
> > **Reviewer:** *4.2 I am not sure NOTEARS, GES, and PC are the most probing methods to compare to. Since this is mostly an algorithmic paper, many more comparisons are needed (SCORE, CAM, NoGAM, RESIT provide good baselines for ANM causal discovery — these are just some of my picks, feel free to change/expand)*
>
> In our initial evaluation, we selected NOTEARS, GES, and PC as they are highly representative and widely adopted baselines that cover three distinct paradigms of causal discovery: score-based (GES), constraint-based (PC), and continuous optimization (NOTEARS), which were discussed in the Related Works section. We have now added comparisons with several traditional causal discovery methods. The results are summarized in **Response Table 5** below, which shows that TACTIC maintains highly competitive performance.
>
> **Response Table 5: Extended baseline comparison**
> |                  | RFF_G  | Linear_U | Chebyshev_G | Sachs  | Syntren |
> | ---------------- | ------ | -------- | ----------- | ------ | ------- |
> | PC               | 61.1 (4.9) | 60.9 (4.7) | 59.8 (6.6)  | 67.1   | 58.1    |
> | GES              | 66.0 (10.6) | 69.0 (10.8) | 59.6 (5.9)  | 61.8   | 36.8    |
> | Notears          | 80.5 (4.0) | 82.0 (4.6) | 52.2 (3.5)  | 61.8   | 49.8    |
> | **RESIT**            | 54.3 (5.4) | 54.1 (5.2) | 49.8 (4.7)  | 62.3   | 64.6    |
> | **GranDAG**          | 58.3 (4.8) | 60.1 (5.9) | 54.1 (4.5)  | 54.2   | 52.5    |
> | **SCORE**            | 86.9 (3.2) | 82.2 (18.7) | 69.2 (7.6)  | 64.9   | 41.0    |
> | **NoGAM**            | 87.6 (2.9) | 79.2 (18.6) | 72.3 (6.4)  | 64.9   | 41.0    |
> | AVICI (scm-v0)   | **97.8 (1.3)** | 75.6 (13.8) | 81.7 (10.5) | 62.3   | 65.4    |
> | TACTIC (random)  | 88.4 (7.0) | 82.3 (7.0) | 79.6 (6.7)  | 58.6   | 72.0    |
> | TACTIC (Notears) | 91.8 (3.1) | **86.3 (4.4)** | **83.0 (8.7)** | **78.9** | **80.1** |
>
> > **Reviewer:** *4.3 Clarity about Table 1*
>
> We will redesign the presentation of Table 1 in section 3.2 and clarify the meaning of "Each specific training data setting is indicated above the result." We will include this in the updated version of the paper before the end of the rebuttal.
>
> > **Reviewer:** *4.4 L222: “AVICI can be considered the strongest model of open source under the SCL paradigm”: any evidence for this?*
>
> We thank the reviewer for this clarification. Our intended meaning was that AVICI is **one of the strongest open-source models under the SCL paradigm**, a claim supported by its established reputation and consistent use as a top-performing baseline in recent literature. For instance, subsequent works like the SiCL model (Zhang et al., 2025) and the BAM model (Froehlich & Koeppl, 2024) benchmark against AVICI as a state-of-the-art reference, where it has demonstrated robust and highly competitive performance, solidifying its status as a leading **publicly** available SCL model. We will revise our manuscript to reflect this more precise characterization.

---

> > ### Comment · Reviewer_V2uH · 2025-11-25
> >
> > I thank the authors for the detailed response. For an easier interaction process on my side, I kindly ask the authors to keep the answer shorter. Below, I’ll respond to the main points.
> >
> > ---
> >
> > > The most fundamental difference: Montagna et al., 2024 focus on the benefit of adding more types of distributions [..] advocating for increased diversity and size in pre-training [..]. In contrast, our Issue 2 exposes a more fundamental combinatorial explosion problem, which cannot be resolved by simply increasing the number of mechanism types or training samples.
> > >
> >
> > Montagna et al. give two contributions
> >
> > 1. They show the issues in OOD settings, due to the mentioned combinatorial issues: they show that unseen mechanisms and noise distributions are detrimental for performance (see e.g. their Figure 2 and all their related discussion). This overlaps with the contributions of this paper
> > 2. They also show, as the authors mention, potential ways to mitigate the issue.
> >
> > So clearly point 1. overlaps with the contribution of the paper, reducing the novelty
> >
> > ---
> >
> > > Note that the scarcity of real-world datasets is one of the key practical problems in causal discovery
> > >
> >
> > Please be aware that, despite this being true, there are many more (pseudo) real-world datasets other than Sachs and syntren. I am not saying that these need to be included: in fact, one of the reasons why there are not a lot of real-world data with the causal graph as a label is that such a label is hard to define. However, if the authors say that they *run extensive experiments on synthetic, pseudo-real, and real data* (written multiple times in the paper), they must run extensive experiments on real data, from different domains, with different graph sizes.
> >
> > In this regard,  thanks to the authors for the additional experiments. In order to understand whether these are representative, a discussion on the datasets must be included. Is this in the updated paper, or to be found somewhere?
> >
> > ---
> >
> > > In reality, no real data perfectly  conforms to the ANM assumption;
> > >
> >
> > This is exactly my point, and in my view, it’s a big limitation of this work. In detail:
> >
> > From a theoretical perspective, identifiability is a combination of assumptions on the source distribution and functional mechanisms, which are used to restrict the class of possible graphs that have generated the data. If you train to map nonlinear ANM datasets to graphs, you are
> > teaching the model to find the causal direction under the assumption of an additive noise model. This is problematic if your real-world data (e.g. Sachs) are not compliant with the ANM assumption, because you are biasing the solution to assume a nonlinear ANM: e.g., look at Example 2 in Montagna et al. (a rephrasing of an example in Hoyer et al., 2009), where the same distribution maps to opposite causal direction based on the assumed SCM class.
> >
> > Then, your methodology relies on foundations that go against the known theory of causality: for me to convincingly support the hypothesis that this is a good idea, there must be much more evidence on a lot real-world data with a lot of variance (e.g. in the graph size, in the domain, …). This is currently not the case, so I can still not advocate for acceptance. I still see the issue.
> >
> > ---
> >
> > > Regarding the response on *reprosucibility issues (4.1.x)*
> > >
> >
> > Thanks for the details. Are the information in this response added to the main paper? If not, can you please do that, and acknowledge that? thanks.

---

> > > ### Author Response · Authors · 2025-11-27
> > > **Clarification: TTT-SCL is a general test-time training framework, not limited by the ANM assumption.**
> > >
> > > > **Reviewer:**  *This is exactly my point, and in my view, it’s a big limitation of this work. In detail:[...] I still see the issue.*
> > >
> > > **Response:**
> > >
> > > First, the reviewer **originally** questioned the value of our method in the ANM setting, and we have clearly demonstrated this value — as shown by its superior performance on the Linear_U setting in **Table 3** and the expanded baselines in **Response Table 5**.
> > >
> > > **Importantly, TTT-SCL does not assume ANM at test time.** ANM is used only as one of the clean and standard identifiable benchmark setting. We apologize that our current background section introduces ANM early, which may unintentionally give the impression that our framework relies on ANM. This is not the intended message, and we will revise the Background and other sections to make this explicit.
> > >
> > > **Our experiments on non-ANM test instances also demonstrate that TTT-SCL is not subject to the ANM assumption.** On real non-ANM data (Sachs), TTT-SCL achieves 78.9 AUROC, outperforming PC (67.1), NOTEARS (61.8), GES (61.8), and AVICI (62.3). On pseudo-real Syntren and non-ANM synthetic mechanisms (Chebyshev), TTT-SCL again achieves the best performance.
> > >
> > > The intuitive reason **why** TTT-SCL achieves good performance in non-ANM scenarios is that **it can obtain empirical distribution information of the test instance** and ensure a one-to-one match between the causal graph and the data of generated training datasets through the structure-induced mechanism (SIM) and resampling. This principle does not contradict Example 2 in Montagna et al., because their research focuses on static SCL, which does not access test instance information during training.

---

> > > > ### Comment · Reviewer_V2uH · 2025-11-27
> > > >
> > > > > This principle does not contradict Example 2 in Montagna et al., because their research focuses on static SCL, which does not access test instance information during training.
> > > >
> > > > Let's assume you are given data generated exactly by the linear ANM in example 2 in Montagna et al. Let's say you run the TTT-SCL procedure twice:
> > > > 1. The first time, you augment the data with linear ANM. Then, you will find the right causal direction
> > > > 2. The second time, you augment the data with a nonlinear ANM. Then, you will find the opposite causal direction.
> > > >
> > > > I hope this clarifies what I mean when I say that the assumption of generating data with an ANM is not neutral, and why I think that there is a flaw in the method if you look at it from a theoretical perspective.

---

> > > > > ### Author Response · Authors · 2025-12-03
> > > > >
> > > > > >Reviewer: The ANM assumption is not neutral.
> > > > >
> > > > > Thank you for your clear explanation of the question. We believe there may be a misunderstanding here that we would like to clarify.
> > > > >
> > > > > Our framework does not assume that the data generation is limited to ANMs. ANM is used only as one of the clean and standard identifiable benchmark setting. Discussions about ANM essentially revolve around identifiability, that is, the selection of the learning target for SCL (Zhang et al. 2025). We further clarify that our TTT-SCL framework is learning target-independent. We explicitly state that the reason we need the identification to DAG assumption is to simplify the learning target for more intuitive comparison with other methods and to understand the core contributions of our TTT-SCL. Even when the learning target is set as DAG in this paper, our method is not limited to ANM; it is applicable to any identification to DAG assumption, including LiNGAM and PNL (**Lines 277-286 in the revised paper**). Furthermore, our method achieved excellent performance regardless of whether the ANM assumption was applied (**Table 2**). Similarly, we also verified the effectiveness of our TTT-SCL framework on the SiCL model with the learning target as skeleton, further demonstrating that the framework is independent of the learning target (**Appendix C**).

---

> > > ### Author Response · Authors · 2025-11-27
> > >
> > > > **Reviewer:**  *Montagna et al. give two contributions[…] So clearly point 1. overlaps with the contribution of the paper, reducing the novelty*
> > >
> > > **Response:** We respectfully **disagree** the reviewer's claim that point 1. overlaps with the contribution of this paper. The distinction lies in the identification of the cause and its implication for the solution.
> > >
> > > **Component vs. Composition**: Montagna et al. attribute failure to **unseen** mechanisms, i.e., lack of diversity, suggesting the solution is training on more types. In contrast, our experiments (**Table 1 & Response Table 3**) reveal the combinatorial failure, i.e., even when the model **has seen** all individual components (mechanisms/noise) during training, it fails on their novel combinations. Moreover, performance does not improve with increasing sample size, which **differs** from the findings of Montagna et al.
> > >
> > > **Implication for TTT**: Therefore, if the problem were only what Montagna described, simply scaling up pre-training data would suffice. However, as we discussed above, covering the Cartesian product of all factors is combinatorially intractable for static pre-training. This specific finding is what theoretically necessitates and justifies our TTT-SCL paradigm over simply improving static SCL.
> > >
> > > Thus, our contribution is not just identifying that SCL fails, but characterizing why static training is insufficient for combinatorial shifts, directly motivating the proposed TTT framework.
> > >
> > >
> > > > **Reviewer:** *In this regard, thanks to the authors for the additional experiments. In order to understand whether these are representative, a discussion on the datasets must be included. Is this in the updated paper, or to be found somewhere?*
> > >
> > > > **Reviewer:**  *Thanks for the details. Are the information in this response added to the main paper? If not, can you please do that, and acknowledge that?*
> > >
> > > **Response:** We are incorporating all the above modifications into the manuscript and will upload the revised version shortly. We will inform you as soon as the updated version is submitted.

---

### Official Review · Reviewer_94dy · 2025-11-03

**Soundness:** 3
**Presentation:** 3
**Contribution:** 4
**Rating:** 8
**Confidence:** 4

**Summary:**

This paper argues that the standard static pre-training paradigm for Supervised Causal Learning (SCL) is flawed, demonstrating its fragility to distribution shifts and failure to generalize from synthetic to real data. It proposes a new paradigm, Test-Time Training for SCL (TTT-SCL), which generates a small, bespoke training set for each test instance on-the-fly. The core idea is to search for training graphs that maximize an "Alignment of Distribution" (AD) score with the test data, penalized by a sparsity term to ensure causal minimality. The paper's implementation, TACTIC, uses stochastic graph refinement to generate this data. Experiments show TACTIC substantially outperforms established SCL and traditional baselines in challenging OOD and real-world settings.

**Strengths:**

The primary contribution is significant and original: it introduces the first test-time adaptation framework for SCL, fundamentally reframing the problem from seeking universal diversity to achieving targeted concentration. Apart from that, I highlight the following points:
- Principled test-time framework for causal discovery; clear link between distributional alignment and causal minimality.
- Rigorous OOD diagnostics (shift, composition, synthetic-real gap).
- Comprehensive experiments with two backbones and multiple metrics.
- Insightful sparsity ablation showing dense graphs mimic data yet harm causal accuracy.

**Weaknesses:**

1) Scalability: results ≤ 20 vars; K = 200 graphs + SIM fitting may be heavy; no runtime or complexity analysis.
2) Initializer dependence: performance drops sharply without NOTEARS seed; robustness unclear.
3) Theory: no guarantees that AD optimization recovers true graph or stable minima.
4) Evaluation breadth: only Sachs as real dataset; no interventional or large benchmarks.

**Questions:**

1) How is the runtime breakdown vs. AVICI for d >= 20, K = 200?
2) How sensitive is λ across datasets; any adaptive tuning needed?
3) How does performance scale with poorer or random seeds?
4) Could hybrid fine-tuning of pretrained SCL with TTT-SCL reduce cost?
5) Is AD robust under misspecified mechanism regressions?
6) Recent works explore adaptive or amortized causal graph generation via generative sampling and refinement (e.g., [1,2]). How does TACTIC relate to these frameworks that also perform dynamic graph generation, and could TTT-SCL benefit from such generative sampling instead of heuristic refinement?

[1] Bayesian Structure Learning with Generative Flow Networks. https://proceedings.mlr.press/v180/deleu22a/deleu22a.pdf
[2] Expert-Aided Causal Discovery of Ancestral Graphs. https://arxiv.org/abs/2309.12032

---

> ### Author Response · Authors · 2025-11-24
> **Response to Reviewer 94dy (1/3)**
>
> Thank you for your recognition of our work, as well as your constructive suggestions and key questions. Please see our point-by-point response below.
>
> >**Reviewer:** *1.K = 200 graphs + SIM fitting may be heavy; no runtime or complexity analysis. How is the runtime breakdown vs. AVICI for d $\ge$ 20, K = 200?*
>
> Causal discovery is fundamentally a scientific inference task where high computational cost is often acceptable in exchange for accurate and reliable results.
>
> **Runtime and scalability analysis.** Concretely, the computational complexity of TACTIC is dominated by the **Stochastic Graph Refinement** step. The search space for Directed Acyclic Graphs (DAGs) with $d$ variables is super-exponential, rendering exhaustive search intractable. Our stochastic search conducts a guided walk through this space, and its complexity is determined by the number of steps $N_{\text{steps}}$ and the cost of evaluating the Alignment of Distribution (AD) score for each candidate graph. The evaluation for a single candidate graph $G$ involves:
>
> -   **Mechanism Fitting**: For each node $X_i$, we fit a causal mechanism $f_i$ (e.g., using a Generalized Additive Model) based on its parent set $Pa_G(X_i)$ from the test data $D_{\text{test}}$ with $n$ samples. Let $k_i = |\mathbf{Pa}_G(X_i)|$ be the in-degree of $X_i$. The cost of fitting for one node is typically $O(n \cdot k_i \cdot l)$, where $l$ is the number of GAM iterations. Critically, due to the sparsity constraint $\text{Sparsity}(G)$ in our joint score function (Eq.~5), which actively penalizes dense graphs, the in-degrees $k_i$ encountered during the search are small. If we let $H$ represent the small, approximately constant maximum in-degree enforced by this constraint, then the cost per node becomes $O(n \cdot H \cdot l)$. Aggregated across all $d$ nodes, the total fitting cost is $O(n \cdot H \cdot d \cdot l)$, which simplifies to $O(n \cdot d)$ since $H$ and $l$ are constants.
> -   **Likelihood Calculation**: After fitting, computing the log-likelihood for all $n$ samples and $d$ variables has a cost of $O(n \cdot d)$.
>
> Thus, the per-step AD evaluation cost is $O(n \cdot d)$. The total complexity of the **Stochastic Graph Refinement** phase is therefore $O(N_{\text{steps}} \cdot n \cdot d)$. The subsequent **Training Data Generation** step involves fitting mechanisms and forward-sampling for only the final $K$ selected graphs, contributing a minor additive term of $O(K \cdot n \cdot d)$, which is negligible since $N_{\text{steps}} \gg K$.
>
> To verify our theoretical analysis above, we present a runtime breakdown for a test instance in **Response Table 1**. The results confirm that the Stochastic Graph Refinement is indeed the time bottleneck, as it involves thousands of AD evaluations. Crucially, the subsequent "Training Data Generation" step, which performs SIM fitting on only the final $K$ selected graphs, constitutes a minor fraction of the total time. Therefore, increasing $K$ has a negligible impact on the overall runtime, as $N_{\text{steps}} \gg K$. ($N_{\text{steps}}=2000, K=200$). We will add this new runtime analysis to the revised manuscript.
>
> **Response Table 1: Runtime breakdown of TACTIC for a test instance**
> |                          | 10 nodes | 20 nodes | 30 nodes |
> | ------------------------ | -------- | -------- | -------- |
> | Stochastic graph refinement | 26 min   | 61 min   | 113 min  |
> | Training data generation (SIM fitting) | 1.3 min | 3.2 min | 5.6 min |
> | Model training            | 3.3 min  | 5.8 min  | 8.3 min  |
>
> **Comparison with AVICI.** It is crucial to recognize the fundamental difference in computational paradigm between TACTIC and pre-trained SCL models. TACTIC operates under a **test-time training** paradigm. For each individual test instance, it performs data generation and model training from scratch. In contrast, a **pre-trained** model like AVICI(scm-v0) follows a pre-training paradigm: it is trained once, extensively, on a massive synthetic dataset, and then applied to all subsequent test instances via a cheap forward pass.
>
> While TACTIC's per-instance cost is higher than a single forward pass of a pretrained AVICI model, it is dramatically lower than the cost of pre-training AVICI itself (which takes hours to days on multiple GPUs, massive training graphs/instances, e.g. $K \\ge 2000$). More importantly, this cost is incurred on-demand to generate a training set explicitly aligned with the specific test instance, guaranteeing robustness without prior distributional assumptions.

---

> ### Author Response · Authors · 2025-11-24
> **Response to Reviewer 94dy (2/3)**
>
> >**Reviewer:** *2. How sensitive is $\lambda$ across datasets; any adaptive tuning needed?*
>
> The choice of $\lambda$ across all our experiments was not arbitrary but was motivated by a theoretical prior. Our score function, $\text{score}(G) = \text{AD}(G, D_{\text{test}}) - \lambda \cdot \text{Sparsity}(G)$, is conceptually aligned with the **Akaike Information Criterion (AIC)**, a well-established model selection criterion formulated as $\text{AIC} = 2k - 2\ln(L)$, where $k$ is the number of parameters and $L$ is the likelihood. In our formulation, $\text{Sparsity}(G)$ is analogous to the number of parameters $k$, and the negative log-likelihood $-\log p(X \mid f)$ is analogous to $-\ln(L)$. The AIC uses a fixed multiplier of 2 for the complexity term, which corresponds to $\lambda = 1$ in our scoring function when AD is implemented via log-likelihood. This provided a principled and fixed starting point.
>
> Guided by this prior, we set $\lambda = 1$ for all datasets in our study. The consistent excellent performance of TACTIC across diverse synthetic, pseudo-real, and real-world benchmarks (as shown in Table~3) itself serves as strong empirical evidence that $\lambda = 1$ is a robust default value that generalizes effectively across different data distributions. We will add $\lambda=1$ in the revised manuscript.
>
> >**Reviewer:** *3. How does performance scale with poorer or random seeds?*
>
> We thank the reviewer for raising this point about the robustness of our method to seed graph quality. We address it from two perspectives:
>
> First, **even with a random DAG as the seed (TACTIC (random)), our method consistently and significantly outperforms traditional causal discovery baselines** like PC and GES across almost all test settings, as shown in Table 3 of the main text. This demonstrates the fundamental robustness of the TTT-SCL framework: even a suboptimal starting point, when guided by our AD and sparsity-driven optimization, can yield a high-quality, aligned training set that leads to superior performance.
>
> Moreover, **the stochastic refinement process demonstrates inherent effectiveness, where the limitation of a poor seed graph can be effectively mitigated by allocating more steps**. As evidence, we empirically compared with a new variant, TACTIC (random-long), which uses a random seed but is allowed a much longer search budget ($N_{\text{steps}}=5000$ instead of the standard $N_{\text{steps}}= 2000$). The results are summarized in **Response Table 2** below.
>
> **Response Table 2: Comparisons of different seed graphs and steps on AUROC**
> |                  | RFF_G | Sachs |
> | ---------------- | ----- | ----- |
> | TACTIC (Notears) | 91.8  | 78.9  |
> | TACTIC (random)  | 88.4  | 58.6  |
> | TACTIC (random_long) | 89.1 | 71.9 |
>
> The results clearly show that with a larger computational budget, the performance gap between a poor (random) seed and a good (Notears) seed narrows considerably. For example, on the challenging real-world Sachs dataset, TACTIC (random-long) achieves an AUROC of 71.9, a significant improvement over TACTIC (random) at 58.6, and moves much closer to the 78.9 achieved by TACTIC (Notears).
>
> This illustrates that the stochastic refinement process is ultimately effective on its own, as a poorer seed can be compensated for with a larger computation budget. This provides users with a flexible trade-off based on their available resources and requirements for per-instance inference time. The use of a strong initializer like Notears is therefore a practical recommendation for efficiency instead of a strict requirement.

---

> ### Author Response · Authors · 2025-11-24
> **Response to Reviewer 94dy (3/3)**
>
> >**Reviewer:** *4. Could hybrid fine-tuning of pretrained SCL with TTT-SCL reduce cost?*
>
> A hybrid approach of fine-tuning a pre-trained model is indeed promising for reducing cost. The integration can be achieved through several methods: fine-tuning the pre-trained SCL model via TTT-SCL, utilizing it as a generator to produce seed graphs, and leveraging its own generated graphs as training data for ongoing self-improvement. We remain open to the hybrid approach and believe that it is a promising research direction.
>
> >**Reviewer:** *5. Is AD robust under misspecified mechanism regressions?*
>
> AD is robust under a certain degree of misspecified mechanism regressions. Our current implementation of the Structure-Induced Mechanism (SIM) step relies on the Generalized Additive Model (GAM) as the default functional approximator. This choice inherently assumes an additive structure between the parental variables, which is a common and often reasonable assumption that balances expressiveness with fitting efficiency.
>
> The result regarding the Chebyshev mechanism confirms that TACTIC provides substantial robustness gains even when the functional form is not perfectly specified. To proactively investigate robustness to mechanism misspecification, we incorporated the **Chebyshev polynomial** mechanism from Froehlich \& Koeppl (2024). This mechanism intentionally includes multiplicative interaction terms that violate the additivity assumed by GAM.
>
> The results in Table~3 are highly informative: while the absolute performance on Chebyshev\_G is lower than on RFF\_G and Linear\_U (as expected due to the model mismatch), TACTIC still **significantly outperforms all baselines**, including the pre-trained AVICI model. This demonstrates that even under some extent of mechanism misspecification, GAM can still achieve a relatively robust approximation. This capability, in turn, ensures that TACTIC maintains its advantage over statically pre-trained SCL models by generating training data that is meaningfully aligned with the test distribution.
>
> Our current results with GAM establish a strong and computationally efficient baseline, showing that TACTIC provides substantial robustness gains even when the functional form is not perfectly specified.
>
> >**Reviewer:** *6. Recent works explore adaptive or amortized causal graph generation via generative sampling and refinement. How does TACTIC relate to these frameworks that also perform dynamic graph generation, and could TTT-SCL benefit from such generative sampling instead of heuristic refinement?*
>
> We thank the reviewers for pointing out the potential connection between generative sampling frameworks like GFlowNets and our TTT-SCL framework.
> Both of them aim to efficiently generate high-quality graph structures from large DAG spaces and employ a sequential decision process to construct the graph (TACTIC through random local modifications, and GFlowNets through stepwise edge addition). However, their goals differ: TACTIC aims to dynamically generate a high-quality training set to train the SCL model, whereas GFlowNets aim to directly learn the posterior distribution of the target graph.
>
> Nonetheless, the TTT-SCL might benefit from generative sampling methods. For example, in TACTIC's "Stochastic Graph Refinement" phase, we can use a trained GFlowNet to generate candidate graphs. It further confirms the generalizability of our proposed TTT-SCL framework, and we leave it as interesting future work.

---

> > ### Comment · Area_Chair_DCti · 2025-11-28
> >
> > Dear Reviewer,
> >
> > Please make sure you read the authors' response and engage with them in the discussion before the end of the discussion period on **Dec 03 '25 09:00 PM UTC**. This is a hard deadline.
> >
> > Thank you for supporting quality peer review at ICLR.
> >
> > AC

---

> > ### Comment · Area_Chair_DCti · 2025-11-28
> >
> > Dear Reviewer,
> >
> > Please make sure you read the authors' response and engage with them in the discussion before the end of the discussion period on **Dec 03 '25 09:00 PM UTC**. This is a hard deadline.
> >
> > Thank you for supporting quality peer review at ICLR.
> >
> > AC

---

### Author Response · Authors · 2025-12-03
**Summary for Area Chairs (1/2)**

We would like to thank AC and all the reviewers for their effort. We have reviewed all reviewers' comments and provided detailed responses to thoroughly address their concerns. We have substantially enhanced the paper with **enriched experiments**, **theoretical clarifications**, and **improved writing**. We have uploaded the **revised version** of the paper, with the changes highlighted in blue. To help Area Chairs understand our interactions with the reviewers and the improvements we've made, we will outline how we addressed the reviewers' questions and their main attitudes toward our paper.

>**Reviewer 94dy**

**Reviewer 94dy highly praised our work and offered many suggestions to make our work more solid, and maintained his high score (score = 8).**

(1) In response to his question about scalability, we have added complexity and runtime analysis showing acceptable cost for causal discovery tasks in **Appendix F** of the revised paper.

(2) Regarding his question about our hyperparameter sensitivity, we clarified that our current value of λ is theoretically based on the prior selection of AIC, and has achieved consistently excellent performance on various datasets (**Table 2**).

(3) Regarding the seed graph initialization issue, we first demonstrate that our method achieves good performance even with random seed graphs (**Table 2, TACTIC (random)**). Furthermore, we conducted additional experiments showing that performance improves further when more steps are run on the seed graph, demonstrating the robustness of our TTT-SCL framework (**Response Table 2**).

(4) To address his concerns about misspecified mechanism regressions, we use Chebyshev as an example to illustrate the robustness of our method.

(5) We also believe that his idea of hybrid fine-tuning of pretrained SCL and generative sampling graph is a very interesting future work that is compatible with TTT-SCL.

> **Reviewer V2uH**

**Reviewer V2uH's main concern was that we did not discuss with the paper Montagna et al., 2024 and the assumptions about ANM.**

(1) The reviewers felt our findings on SCL limitation were similar to those of Montagna et al. 2024, significantly reducing our novelty. However, our core findings and derived solutions are **totally different** from that paper. We argue that the failure of the combined generalization (graph*mechansim*noise) is the root cause of the pretrain SCL generalization failure, leading to the TTT-SCL solution that dynamically generates training data based on test data (**Figure 2**). Montagna, on the other hand, only observed that SCL performance degraded on unseen mechanism/noise distributions, thus recommending increasing the diversity of training data, which is the exact opposite of our approach. We have added a discussion of this work in the Related Works section of our paper (**Lines 510-520**).

(2) The reviewers felt our method was limited by the ANM assumption. Discussions about ANM essentially revolve around identifiability, that is, the selection of the learning target. Firstly, we further clarify that our TTT-SCL framework is learning target-independent. We explicitly state that the reason we need the identification to DAG assumption is to simplify the learning target for more intuitive comparison with other methods and to understand the core contributions of our TTT-SCL. (**Lines 277-286**). Furthermore, our method achieved excellent performance regardless of whether the ANM assumption was applied (**Table 2**). Similarly, we also verified the effectiveness of our TTT-SCL framework on the SiCL model with the learning target as skeleton, further demonstrating that the framework is independent of the learning target (**Appendix C**).

(3) The reviewers suggested we add more baseline comparisons and pseudo-real data comparisons. We selected four representative causal graphs from bnlearn as pseudo-real data to further demonstrate the superior performance of TTT-SCL (**Appendix G**). We also added comparisons with SCORE, RESIT, and NOGAM methods (**Table 2**).

> **Reviewer Dm4H**

**Reviewer Dm4H generally approved of our work, hoped we could clarify some technical issues and expressed a willingness to improve the score.**

(1) Code availability: We have released code and provided usage guidelines to ensure the reproducibility of our work. The link is https://anonymous.4open.science/r/TACTIC-B1B0/.

(2) We clarified our method for training SCL models from scratch for each instance.

(3) We have emphasized that TTT-SCL generates a training set rather than outputting graphs directly, thus clarifying that we represent a different paradigm from score-based methods. Furthermore, we have demonstrated through additional experiments that supervised learning based on the training set brings additional performance improvements (**Table 4**).

(4) We clarified that we have confirmed the issue of compositional generalization and improved **Figure 2** to make the conclusions more obvious.

---

### Author Response · Authors · 2025-12-03
**Summary for Area Chairs (2/2)**

> **Reviewer mjjp**

**Reviewer mjjp raised higher requirements from a theoretical perspective and offered suggestions on the writing; however, they felt that our work was innovative enough, so the score was raised to a positive score (4 -> 6).**

(1) We further clarify that our TTT-SCL framework is learning target-independent. We explicitly state that the reason we need the identification to DAG assumption is to simplify the learning target for more intuitive comparison with other methods and to understand the core contributions of our TTT-SCL. (**Lines 277-286 of the paper**)

(2) Regarding the reviewers' concerns about overfitting in the parametric regression step, firstly, we have sparsity constraints on the graph; secondly, we use GAM, a relatively simple parametric fitter, to avoid using overly complex methods such as neural networks, thus further avoiding the risk of overfitting.

(3) The reviewers believe that other score-based methods can serve as the source of the seed graph in our TACTIC method, which perfectly demonstrates the flexibility of our framework, which can be combined with traditional methods through the seed graph as an interface.

(4) To significantly improve the quality of the paper, we restructured **Section 1 (Introduction)** and **Section 3.1 (Experiment Setup)** to enhance overall readability. We also reviewed all abbreviations and formulas used throughout the paper. The original Table 1 has been redesigned and is now **Figure 2**.

*We emphasize that this work is a preliminary exploration aimed at filling the unexplored area of how SCL should address generalization. By identifying the fundamental limitations in current SCL practice, we demonstrate the necessity of a paradigm shift from static pre-training to instance-aligned training. By introducing the idea of test-time training into SCL, TTT-SCL can be viewed as a foundational framework—demonstrating the viability of this shift and opening the path for future work.*

---

### Meta-Review · Area_Chair_HZCq · 2026-01-06

**Summary:**

This manuscript proposes a test time training framework, which searches for a small set of candidate graphs that best match the test distribution, synthesizes paired graph data by fitting mechanisms to the test data, and then fine tunes an existing SCL model.

The reviews are pretty split. One reviewer is quite enthusiastic and sees this as a genuinely new and principled direction, with strong OOD diagnostics and solid experiments. Others are much less convinced and sit around borderline reject, with one clear reject. The positives are fairly consistent: the problem is important, the failure cases are well demonstrated, and the test time adaptation angle is interesting and could be impactful if it holds up. But the main weaknesses also recur. A couple reviewers feel the novelty and positioning are not tight enough, especially relative to prior papers that already discuss OOD fragility for SCL, and the paper does not really spell out what is new vs what is re confirming known issues. More importantly, the method leans on fairly strong assumptions and ends up looking quite close to just do causal discovery with a search score, which makes the framing as a new SCL paradigm less convincing. On top of that, there is limited evidence on scalability and compute cost, sensitivity to initialization seems nontrivial, and the real world evaluation is still narrow. Reproducibility details and code availability also came up.

After reading the paper and the reviews, I recommend rejection. The core idea is promising, but the story does not fully hold together. It is not clear that this is a robust, general improvement over strong classical causal discovery baselines under the same assumptions, and the empirical case is not broad enough to justify the big claims. With clearer positioning, stronger baselines, and better evidence on robustness and efficiency, this could become competitive, but I do not think it is there yet.

**Reviewer Concerns:**

The rebuttal did a decent job addressing the search-related comments. In particular, they clarified the search space question and gave more concrete discussion around the search strategy and efficiency, and they also responded on the reproducibility. The assumption of generating data with an ANM is still outstanding.

**Reviewer Scores:**

Reviewers 94dy and Dm4H did not response the rebuttal. So, I think that they will not change their scores.

Reviewer mjjp stated that his/her concerns are partly addressed, and he/she has raised my score accordingly.

Reviewer V2uH still seems stuck on the core assumption. So I think his/her score would basically stay the same.

---

### Decision · Program_Chairs · 2026-01-26

Reject